# Biology of the mRNA Splicing Machinery and Its Dysregulation in Cancer Providing Therapeutic Opportunities

**DOI:** 10.3390/ijms22105110

**Published:** 2021-05-12

**Authors:** Maxime Blijlevens, Jing Li, Victor W. van Beusechem

**Affiliations:** Medical Oncology, Amsterdam UMC, Cancer Center Amsterdam, Vrije Universiteit Amsterdam, de Boelelaan 1117, 1081 HV Amsterdam, The Netherlands; m.blijlevens@amsterdamumc.nl (M.B.); j.li2@amsterdamumc.nl (J.L.)

**Keywords:** alternative splicing, splicing dysregulation, splicing factors, NSCLC

## Abstract

Dysregulation of messenger RNA (mRNA) processing—in particular mRNA splicing—is a hallmark of cancer. Compared to normal cells, cancer cells frequently present aberrant mRNA splicing, which promotes cancer progression and treatment resistance. This hallmark provides opportunities for developing new targeted cancer treatments. Splicing of precursor mRNA into mature mRNA is executed by a dynamic complex of proteins and small RNAs called the spliceosome. Spliceosomes are part of the supraspliceosome, a macromolecular structure where all co-transcriptional mRNA processing activities in the cell nucleus are coordinated. Here we review the biology of the mRNA splicing machinery in the context of other mRNA processing activities in the supraspliceosome and present current knowledge of its dysregulation in lung cancer. In addition, we review investigations to discover therapeutic targets in the spliceosome and give an overview of inhibitors and modulators of the mRNA splicing process identified so far. Together, this provides insight into the value of targeting the spliceosome as a possible new treatment for lung cancer.

## 1. Introduction

A central dogma of eukaryotic cell biology describes the transcription of genes from DNA in the cell nucleus into messenger RNA (mRNA), which is transported to the cytosol where it is translated into proteins. Hence, mRNA represents the crucial intermediate between “the script” of the cell or its genetic code and “the workers” of the cell, the proteins. In 1977, Nobel laureates Phillip Sharp and Richard Roberts changed our view on how genetic codes are used to produce proteins. They discovered that when hybridized, the major late mRNA transcript from an adenovirus and its DNA template did not form a continuous hybrid DNA-RNA double strand. Instead, electron microscope analysis visualized single strand loops of DNA protruding from different sections of the hybrid double strand [1,2]. Their important finding showed that the genetic message on the DNA is discontinuous and suggested that in eukaryotes the transfer of genetic information from DNA to mRNA is not just by transcription, but also includes removal of at that time considered “invalid” genetic information from a pre-mRNA intermediate transcript. The genetic information without translation function was called introns, while the genetic information that is retained in the mRNA and is subsequently translated by ribosomes into protein was called exons. We now know that the mRNA splicing process, in which introns are removed from the primary precursor transcript to yield mature mRNA, takes place in a large complex of proteins and small RNA molecules called the spliceosome. In recent years, spliceosomes in the process of catalyzing mRNA splicing have been captured and analyzed successfully [3,4,5,6], providing the possibility for in-depth study of the splicing process and splicing disorders. The mRNA splicing process includes highly dynamic spliceosome formation, rearrangement and catalytic steps, which are discussed below.

The mRNA splicing machinery starts to assemble onto pre-mRNA while it is being transcribed in the nucleus [7]. Regardless of the number of introns in a pre-mRNA, four spliceosomes assemble with the pre-mRNA in a complex coined the supraspliceosome, allowing coordinated simultaneous splicing of four introns [8,9]. Emerging evidence has shown that the supraspliceosome, in addition to the mRNA splicing machinery, also harbors other mRNA processing activities [10,11]. Many of the proteins that carry out or initiate these reactions are recruited to the pre-mRNA through interaction with the carboxy-terminal domain of RNA polymerase II [7], allowing immediate processing of the nascent pre-mRNA. Since RNA processing reactions are all enzymatic activities, with kinetics and efficacies dependent on the conditions, they influence each other. The vast majority of mRNA splicing occurs co-transcriptionally, but the process can continue after release of the transcript from its DNA template to complete splicing of—in particular—the last transcribed 3′ introns [12]. The supraspliceosome is considered the master coordinator of quality-controlled mRNA processing in the cell nucleus, producing correctly spliced mature mRNA for subsequent export to the cytosol [13].

The boundaries between exons and introns on the pre-mRNA are delineated by splice (donor and acceptor) sites. When these sites are effectively recognized by the spliceosome, it completely removes the introns and anneals the adjacent exons to yield the mature mRNA. The efficacy at which splice sites are recognized by the spliceosome is, however, not absolute. In contrast, the utilization of splice donor and acceptor sites by the mRNA splicing machinery is context dependent [14,15]. As will be discussed in detail later, splice site utilization is regulated by cis-acting RNA elements and trans-acting proteins. Consequently, many single pre-mRNAs can be processed into multiple mature mRNAs carrying joined exon combinations that encode different protein variants, often with distinct functions. The vast majority of transcribed human genes are subject to this so-called alternative splicing (AS) [16]. On average, human gene transcripts are processed into three or more alternatively spliced mature mRNA transcripts [17]. This contributes to protein diversity and facilitates cell differentiation and tissue development [18,19,20].

The functional impact of AS requires for it to be tightly regulated and controlled. Hence, not surprisingly, dysregulated mRNA processing, such as mRNA splicing, is an important cause of disease, including cancer [21]. A pan-cancer analysis [22] showed that tumors harbor up to 30% more AS events than normal tissues, with many tumors harboring thousands of AS events that are not detectable in normal samples. As will be discussed herein, abnormal splicing of tumor-related genes is associated with cancer development and progression, as well as with resistance to therapy. In this review, we first discuss the structure and function of the spliceosome, the mRNA splicing reaction and its regulation. In view of the intimate interaction with other co-transcriptional RNA processing activities, we do this in the context of the supraspliceosome. Next, we review current knowledge of cancer-related splicing dysregulation contributing to carcinogenesis, progression and treatment resistance, with a focus on lung cancer. Finally, we discuss potential therapeutic interventions targeting the spliceosome that are considered for cancer treatment. Concerning the latter, we devote particular attention to core components of the spliceosome that have until recently been overlooked as potential therapeutic targets.

## 2. Structure and Function of the (Supra) Spliceosome

The supraspliceosome comprises a single pre-mRNA in association with four spliceosomes [9,23]. The RNA splicing machinery, splicing reaction and alternative splicing are described in more detail in Section 2.1. The presence of intact pre-mRNA is essential for the integrity of the supraspliceosome [8]. A “rolling model” was proposed in which the pre-mRNA “rolls” through the supraspliceosome, thereby making new introns available to the splicing machinery. The supraspliceosome is co-transcriptionally assembled and does not disassemble during the splicing reaction. Its composition remains intact as snRNA:pre-mRNA basepairing and protein:RNA interactions are reformed on each subsequent intron [24]. Electron microscopic studies of the structure of the supraspliceosome revealed that the four native spliceosomes consist of a large and a small subunit, of which the first faces towards the outside and the latter faces the inside of the supraspliceosome, forming contacts with the neighboring spliceosomes. The small subunit contains a hole that leads into the cavity between its neighboring small and large subunits, presumably to facilitate the pre-mRNA to connect all four spliceosomes [25]. An in-silico study proposed that the snRNPs reside within the large subunit, leaving the non-snRNPs to be attributed to the small subunit [26]. A schematic depiction of the supraspliceosome is given in Figure 1a.

### 2.1. The RNA Splicing Machinery and the Splicing Reaction

A core task of the supraspliceosome is catalyzing the mRNA splicing reaction in its four spliceosomes. The splicing reaction itself consists of two subsequent transesterification reactions that together yield two ligated exons and excise the intronic sequence. The splicing process is initiated by recognition of splice sites (ss) in intronic sequences. The vast majority of introns in eukaryotes are U2-type introns that contain highly conserved GT and AG dinucleotides at their 5′ss and 3′ss, respectively. The remaining part of the introns are U12-type introns, accounting for ~0.01-0.02% of eukaryotic introns, that have AT-AC boundary sequences [27]. Although there are some exceptions, U2-type introns undergo so-called canonical splicing by the major spliceosome, whereas U12-type introns are spliced by the minor spliceosome through non-canonical splicing. As the majority of human introns are U2-type introns, this review will only focus on the major spliceosome. The major spliceosome consists of the five snRNPs U1, U2, U4, U5 and U6; DExD/H-type RNA-dependent ATPases/helicases that facilitate structural remodeling of the snRNPs at different steps in the splicing process, as well as many other splicing factors that regulate splice site usage and the mRNA splicing reaction. The snRNPs are the core units of the spliceosome. In this section, each snRNP’s structure and function will be described in more detail. An overview of all the snRNP-specific and -associated RNA splicing factors according to the KEGG, amiGO and Reactome databases is given in Appendix A and the most important ones are shown in Figure 1b.

#### 2.1.1. Biogenesis of the Spliceosome: Assembly and Transport of Sm-snRNA Complexes

Biogenesis of snRNPs for assembly of spliceosomes has been reviewed more extensively elsewhere [28,29,30,31] and is illustrated in Figure 2. The U1, U2, U4 and U5 snRNAs are transcribed in the nucleus and exported to the cytoplasm, where they associate with the seven Sm proteins that form a doughnut-shaped circular structure around the snRNA. In the case of the U6 snRNA, the LSm proteins form the typical heptameric ring structure around its Sm site and its biogenesis presumably occurs within the nucleus. The important (L)Sm-snRNA structure in the heart of the spliceosome is discussed in detail below. Complete Sm rings incorporating their snRNA (termed Sm-snRNA complexes here) need to translocate back to the nucleus where they reside in Cajal bodies. There, further snRNP biogenesis takes place, by binding of U snRNP particle-specific proteins to the Sm-snRNA complexes. Mature snRNPs are primarily located in nuclear membrane-less organelles termed nuclear speckles that serve as a reservoir for spliceosome components. Nuclear speckles are found in interchromatin regions close to actively transcribed genes (reviewed by [32,33]). Many proteins involved in transcription, epigenetic regulation, and RNA processing, modification and packaging are present in these speckles, making them nuclear gene expression hubs. The localization of proteins in nuclear speckles is regulated by post-translational modifications such as phosphorylation, addition of phosphoinositol derivates, ubiquitination and SUMOylation [32]. The localization in speckles is important, as increasing the levels of U1 snRNPs did not achieve enhancement of mRNA production when no speckles were present [34]. For use in the mRNA splicing reaction, snRNPs are recruited from the speckles to active transcription sites at the interface of speckles and chromatin. The Sm and LSm proteins have been postulated to be the earliest spliceosomal components, with their gene family nearly achieving its current composition already by the time the last eukaryotic common ancestor emerged approximately two and a half billion years ago [35]. The seven Sm proteins that together form the heteroheptameric ring structure are denoted SmB/B’, SmE, SmF, SmG, SmD1, SmD2 and SmD3. Additionally, SmN is a tissue-specific substitute for SmB/B’ expressed primarily in the brain and heart that affects mRNA splicing through downregulation of mature U2 snRNP when it is incorporated in its Sm ring [36]. The structurally highly similar (L)Sm proteins form a heptameric ring structure around each snRNA, which presumably functions as a platform for other snRNP proteins to assemble onto. Sm proteins are crucial for the assembly, stability and nuclear import of snRNPs and hence for proper functioning of the spliceosome.

Each Sm protein consists of a short N-terminal α helix and five anti-parallel β strands, representing the highly conserved Sm fold. β strands 1–3 represent Sm motif 1 which is involved in the protein-snRNA interaction. β4 and β5 represent Sm motif 2 which is involved in the protein–protein interactions within the Sm heptameric ring structure. β4 interacts with the β5 of its neighbor through the formation of hydrogen bonds. The Sm motifs are highly conserved [37]. The Sm ring structure is further stabilized by hydrophobic residues that point towards its center and make contacts with other Sm proteins [38].

The association of the Sm ring with the U snRNA represents the first phase of the snRNP assembly [39]. The stepwise assembly of the Sm-snRNA complex takes place in the cytoplasm (Figure 2) and is mediated by the methylsome complex (consisting of protein arginine N-methyltransferase 5 (PRMT5), methylosome protein 50 (WDR77) and methylosome subunit pICln) and the SMN complex (consisting of Gem-associated proteins Gemin 2–8 and SMN1 and 2) [40,41]. SmD1-SmD2 initially forms a dimer bound by pICln that is assembled onto PRMT5/WDR77 where SmD1 is symmetrically dimethylated. SmD3/SmB/B’, in parallel, is bound by another pICln molecule and recruited to a second PRMT5/WDR77 complex where SmD3 and SmB are both symmetrically dimethylated. In plants, Sm protein dimethylation by PRMT5 was shown to be required for recruitment of the NineTeen/Prp19 complex (NTC) and therefore proper functioning of the spliceosome [42]. In addition to PRMT5, PRMT7 was also found to be required for symmetrically dimethylating Sm proteins [43]. Subsequently, the SmF-SmE-SmG trimer binds to SmD1-SmD2-pICln on the methylosome complex and forms the 6S complex [40]. The 6S complex is an Sm ring intermediate in which pICln functions as an Sm protein mimic reserving space for SmD3-SmB/B’ [40,44]. Of note, pICln was postulated to not only function as structural chaperone in the formation of the Sm protein ring, but also to prevent the formation of aggregates by unassembled Sm proteins [41]. In the final steps of the Sm ring assembly, the 6S complex and SmD3-SmB/B’ complex release their pICln subunits and are loaded onto the SMN complex through interactions with Gemin2, where the final heptameric Sm ring is formed [44,45]. Interestingly, Sm proteins are not the only proteins that adopt the Sm fold, as this structural arrangement was also observed for Gemin6 and Gemin7 in the SMN complex. Gemin6 and Gemin7 form a dimer through their β4 and β5 strands, respectively. The β5 of Gemin6 and β4 of Gemin7 are involved in binding Sm proteins, thereby facilitating the interaction between Sm proteins and the SMN complex [46]. Defects in the association of Sm proteins with the SMN complex can have drastic phenotypic consequences, as malfunction of the SMN complex due to loss of SMN1 leads to spinal muscular atrophy (reviewed by [47]). Additionally, the F22S mutation in SmE leads to the disruption of its interaction with the SMN complex and is associated with microcephaly [48]. Above-mentioned SMN defects give rise to aberrant splicing patterns.

Gemin5 is an RNA-binding protein (RBP) that is part of the SMN complex and is involved in the recruitment of the snRNA towards the Sm proteins through recognition of the Sm site and the 7-methylguanylate (m^7^G) cap of the snRNA [49,50,51,52]. The Sm site of U1 has a weaker binding affinity than the Sm sites of other U snRNAs for Gemin5. Here, U1 snRNP 70 kDa (SNRNP70) in the cytoplasm recruits Gemin2-Sm complexes directly to U1 snRNA, independent of Gemin5 [53]. The Sm proteins are organized around the snRNA in the following consecutive order: SmE, SmG, SmD3, SmB, SmD1, SmD2 and SmF, with each Sm protein directly interacting with a single nucleotide of the Sm site [54]. The Sm site consists of a 5′ adenosine preceding five consecutive uridines (with U1 being the exception that contains a guanine instead of the fourth uridine), followed by a guanine (with U5 and U6 snRNAs being the exceptions, which contain a uridine or a 2′,3′-cyclicphosphate group, respectively) [54,55,56,57]. The 5′ adenosine and the 2′-OH groups of the sugar backbone of the Sm site are important for stable Sm-snRNA complex formation. Additionally, flanking nucleotides of the Sm site determine the rate of Sm protein assembly onto the snRNA [39]. SmE, SmF and SmG make the initial contact with the Sm site, which is stabilized by SmD1 and SmD2, of which the contact between SmG and the first uridine is highly conserved [57]. Every Sm protein clamps its corresponding Sm site nucleotide in between their loops L3 and L5 [38,54]. The positioning of the Sm ring relative to the Sm site is not the same for every U snRNA. In the human U1 snRNA, the last two nucleotides of the Sm site interact with SmD2 and SmF [54], whereas SmD1 and SmD2 interact with the last two nucleotides of the U4 snRNA Sm site [38]. A study in yeast demonstrated that mutagenesis of conserved Sm motif 1 amino acids in individual Sm proteins did not compromise cell viability, but simultaneous mutation in two Sm proteins was lethal [58]. Thus, at least 6 intact RNA binding sites in the heptameric Sm ring seem sufficient for Sm-snRNA complex formation. During the assembly of the Sm ring around the snRNA by the SMN complex, the snRNA cap is hypermethylated by trimethylguanosine synthase 1 (TGS1) into 2,2,7-trimethylguanosine (m3G) [59].

The m3G-cap is recognized by snurportin-1, which together with importin-β [60] transports the fully assembled Sm-snRNA core complexes back into the nucleus where they are loaded into membrane-less nuclear Cajal bodies. Here, the additional snRNP-specific proteins are loaded onto Sm-snRNA core complexes and snRNP maturation takes place [61]. snRNAs are 2′-O-methylated by small Cajal body-specific RNAs (scaRNAs) and pseudouridylated [28,29,31]. Additionally, snRNAs can be methylated at the N6-position of 2′-O-methylated adenosine residues, which affect spliceosomal function. For instance, the U6 snRNA is methylated at its 2′-O-methyl adenosine at position 43 by RNA N6-adenosine-methyltransferase METTL16 and this affects 5′ss recognition [62]. In the U2 snRNA, the adenosine at position 30 is 2′-O-methylated and subsequently N6-methylated by N(6)-adenine-specific methyltransferase METTL4, which affected 3′ss usage [63]. Both N6-methylated and 2′-O-methylated forms can coexist, as the N6-methylation can be removed by the RNA demethylase alpha-ketoglutarate-dependent dioxygenase FTO [64]. As the adenosines are 2-O′-methylated before they are N6-methylated, it is assumed that the latter also occurs in Cajal bodies upon re-uptake of the Sm-snRNA complex into the nucleus.

#### 2.1.2. Biogenesis of the Spliceosome: Structure and Assembly of the U1 snRNP

The different snRNAs incorporated in the Sm-snRNA complexes direct the formation of the different U snRNPs of the spliceosome in the Cajal bodies (Figure 1b and Figure 2). The U1 snRNA forms four stem-loop (SL) structures, which are oriented in a latin cross-like shape, with SL4 representing the stem. The Sm site is located between the stem and the four-helix junction [65]. The U1 snRNA four-helix junction is situated over a flat surface consisting of the N-termini of each Sm protein. The N-terminus of SmD2 is particularly long and extends into the minor groove of the U1 snRNA. SmB interacts with the SL2 backbone [65]. SL1 and SL2 are also bound by the SNRNP70 and U1 snRNP A (SNRPA) proteins, and SNRNP70 helps to guide the snRNA through the cavity of the Sm ring, together with SmD1 and SmD2 [54]. The U1 Sm core has an additional and unique assembly pathway, in which SNRNP70 plays the key role. As mentioned above, it recruits Gemin2-Sm complexes directly to U1 snRNA, independent of Gemin5. Moreover, SNRNP70 inhibits the formation of other snRNP Sm cores, thereby acting as a regulator of the cell’s snRNP repository. This extra U1 Sm core assembly pathway could be an explanation as to why the U1 snRNP is the most abundant snRNP in vertebrates [53]. Another protein involved in U1 Sm core assembly is the RBP FUS, which associates with U1-related proteins and SMN complexes. Mutations in FUS that are associated with amyotrophic lateral sclerosis (ALS) were found to dysregulate SMN function, leading to loss of snRNA levels and affected splicing patterns [66].

The U1 snRNP is the first snRNP to be recruited to pre-mRNA to start the splicing reaction. The composition of U1 snRNP is shown in Figure 1b. A cryo-EM study of the yeast U1 snRNP revealed a shape similar to that of a footprint [67]. The U1 snRNP core (or the foot’s ball) is composed of the Sm-snRNA complex and the SNRNP70, SNRPA and SNRPC proteins, with SNRNP70 and the stem-loop 1 (SL1) and SL3 of the U1 snRNA sticking out like toes. The auxiliary area (or the foot’s heel) consists of Prp42, Luc7/LUC7L2, Snu56, Nam8/TIA-1 and Prp39/PRPF39. There is no human homologue reported of the yeast Prp42. Instead, in humans, PRPF39 forms a homodimer which interacts with SNRPC, connecting the ball with the heel of the foot, and mimicking the Prp39/Prp42 heterodimer observed in yeast. In immunoprecipitation [68] and X-ray crystallography [65,69] studies of the human U1 snRNP, it was revealed that the N-terminal domain of SNRNP70 plays a crucial role in holding the core domain of the U1 snRNP together, specifically through interacting with SmD2 [65,68,69] and SmB/B’ [68]. Moreover, A feedback regulatory mechanism has been described between SNRNP70 and SNRPC, effectuating efficient U1 snRNP homeostasis. Specifically, SNRPC promotes alternative splicing of the SNRNP70 transcript through usage of an alternative 3′ss. This introduces a premature termination codon (PTC), resulting in a truncated splice variant of SNRNP70 that is targeted for nonsense-mediated decay (NMD), and therefore in decreased protein expression of SNRNP70. This in turn leads to decreased incorporation of SNRPC in the U1 snRNP, restoring proper splicing of SNRNP70 to produce the functional protein [70].

#### 2.1.3. Biogenesis of the Spliceosome: Structure and Assembly of the Other snRNPs

The U2 snRNP, which is the next spliceosome unit to be recruited, consists of the U2 Sm-snRNA complex, the SF3a and SF3b complexes and additional U2-specific and -related proteins. The U2 snRNA, like U1 snRNA, forms four stem-loops in the Sm-snRNA complex, of which SL3 and SL4 are bound by U2 snRNP A′ (U2-A′) and B″ (U2-B″) proteins. SL2a is contacted by the SF3b complex. The branchpoint-interacting stem-loop (BSL) is located between SL2a and SL1, and is clamped between SF3B1’s HEAT domains. SF3A3 contacts the base of the BSL. The Sm site is located between SL2b and SL3 [71]. The SF3a complex consists of splicing factor 3A subunits 1, 2 and 3 (SF3A1, SF3A2 and SF3A3). SF3A1 facilitates the interaction between the U1 and U2 snRNPs by interacting with the U1 snRNA through its ubiquitin-like (UBL) domain [72]. SF3A1, but also U2-related Calcium Homeostasis Endoplasmic Reticulum Protein plays an additional role in the recruitment of U2 snRNP towards the pre-mRNA through their interactions with branchpoint-bridging protein (BBP)/Splicing Factor 1 (SF1) [73]. The SF3a complex bridges the Sm-snRNA complex with the SF3b complex [71]. The SF3b complex consists of RNA splicing factor 3B subunits 1, 2, 3, 4, 5 and 6 (SF3B1-6) and PHD finger-like domain-containing protein 5A (PHF5A). In this complex, SF3B6 is positioned in such a way that it can bind to the branchpoint sequence (BPS), facilitating BPS recognition by the U2 snRNP. SF3B1 adopts a closed conformation surrounding SF3B6, and serves as a platform for BPS binding together with PHF5A [71,74]. SF3B1 also appears to play a role in guiding the U2 snRNP towards the pre-mRNA, as this mRNA splicing factor was shown to interact with chromatin at nucleosomes located at exons to be spliced [75].

While U1 and U2 snRNPs assemble individually before being recruited in the spliceosome to participate in the RNA splicing reaction, U4, U5 and U6 preform a tri-snRNP complex in two steps, where U4/U6 first assemble as di-snRNP before the U5 snRNP attaches. U5 snRNA contains one large stem-loop (SL1) and a smaller SL2. The Sm site is located between these loops [76] at the 3′ end [4]. In the mRNA splicing process, SL1 is important for basepairing with the 5′ exon in the pre-mRNA [77]. In yeast, the U5 Sm core serves as a protein-binding platform for U5 specific proteins Prp8/PRPF8 and Snu114/EFTUD2 that associate either through direct interaction with U5 snRNA or with the Sm ring, respectively [4]. The U4 and U6 snRNAs are different from U1, U2 and U5, as these are duplexed within the U4/U6 di-snRNP and U4/U6.U5 tri-snRNP. The U4 snRNA comprises three SLs. The Sm site is located at the 3′ end [78] and is flanked by SL2 above the flat face of the Sm ring, and by SL3 below the tapered side of the Sm ring. The α-helix that makes up the long N-terminus of SmD2 interacts with SL2 and its lysine-rich L4 loop between β3 and β4 interacts with the backbone of SL3 of the U4 snRNA. Moreover, SL2 interacts with SmB and SmG, and SL3 interacts with all Sm proteins except SmG and SmD3 [38]. On either side of SL2, stem 1 and 2 are basepaired with the U6 snRNA [4,78]. The U4/U6.U5 tri-snRNP is cone-shaped, with the U5 snRNP core located at the tip and the U4/U6 di-snRNP in the broader top part, with the (L)Sm heptamers located at the outer corners of the cone. The following proteins are involved in the U4/U6.U5 snRNP assembly. Small Nuclear Ribonucleoprotein 13 (SNU13) binds to a stem-loop in the U4 snRNA duplexed with the U6 snRNA. Next, pre-mRNA Processing Factors (PRPF) 31, 3 and 4 are recruited, giving rise to the complete U4/U6 di-snRNP [79]. Prior to U5 snRNP assembly, PRPF8, EFTUD2 and SNRNP200 form an assembly intermediate with protein AAR2 homolog. The actual assembly of the U5 snRNP is supported by heat shock protein 90 and R2TP complex (consisting of RuvB-like 1, RuvB-like 2, PIH1 domain-containing protein 1 and Homeobox-containing protein 1) and zinc finger HIT domain-containing protein 2 [80,81].

The U6 snRNA is the exception from all other snRNAs in the sense that it is not bound by Sm, but by LSm proteins. It does so at its 3′ end, where it is uridylated by Terminal Uridylyl Transferase 1 [82]. The LSm ring can only recognize U6 snRNA (and not the other snRNAs) because this is the only snRNA that contains the 3′-terminal U tract [83,84]. In yeast, U6 snRNA’s 3′ end reaches into the ring structure, but does not stick through it as observed for the other snRNAs, thereby only interacting with one side of the ring [85]. The authors speculate that this leaves RNA-binding domains on the other side of the ring accessible to facilitate interactions between the U4 and U6 snRNAs [85]. Indeed, the LSm proteins were shown to facilitate the formation of the U4/U6 duplex [83]. LSm proteins share homology with Sm proteins [83], also form the Sm fold consisting of a short N-terminal α helix and five anti-parallel β strands [86] and similar to the Sm ring also assemble in a stepwise manner. LSm6-LSm5-LSm7 resembles SmF-SmE-SmG but at least in yeast forms a hexameric LSm657-657 intermediate, which subsequently incorporates LSm2-LSm3 (resembling SmD1-SmD2) and finally LSm4-LSm8 (resembling SmD3-SmB/B’), to form the nuclear LSm2–8 complex that is incorporated in the U6 snRNP [86]. Similar to SmD3 in the Sm ring, LSm4 is symmetrically dimethylated which enables interaction with the SMN complex [87]. Comparable to the Sm ring, for each LSm protein, its β4 strand interacts with the β5 strand of the neighboring LSm protein, and the LSm ring is stabilized by hydrophobic interactions through N-terminal α helices [86].

#### 2.1.4. Dynamic Composition of the Spliceosome: Assembly on the pre-mRNA Substrate

Throughout the RNA splicing cycle, different spliceosome intermediates are formed, termed the E (early), A (pre-spliceosome), B (pre-catalytic), B^act^ (activated), B* (catalytically activated; for the first transesterification reaction), C (catalytic), C* (catalytically activated; for the second transesterification reaction) and P (post-splicing) complexes (Figure 3). These intermediates correspond to specific phases of the splicing process, and consist of varying compositions of snRNPs and splicing factors that are described in more detail below. Hence, during the splicing reaction, snRNPs and splicing factors are recruited, rearranged and released in a sequential manner, making the spliceosome a highly dynamic and fluid structure. Many papers have been published over the past decade regarding the yeast and human spliceosome intermediates, describing their structural properties and protein and RNA components (reviewed by [88,89,90]), the major findings of which are summarized here. Notably, the genomic architecture in lower eukaryotes such as yeast is different from that in higher eukaryotes such as mammals. The former usually have relatively long exons and short introns; the latter often short exons and sometimes very long introns. Most fundamental studies into the biology of the spliceosome were done in yeast, or using recombinant transcripts with short introns. Therefore, the general description of spliceosome assembly below primarily applies to pre-mRNAs with short introns, known as the intron definition model. The steps in the process that are probably different for transcripts with long introns, according to the postulated exon definition model [91], are mentioned separately.

The early complex E is the first intermediate that can be discerned in the pre-mRNA splicing process. As mentioned above, the intronic sequence that is to be spliced out contains the highly conserved dinucleotides GT and AG at the 5′ss and 3′ss, respectively, that are recognized by the spliceosome. Moreover, the BPS and polypyrimidine tract (PPT) in the intron play crucial roles in the recruitment of splicing factors. The complex E intermediate is formed when U1 snRNP binds to the 5′ss through basepairing with the 5′ end of its U1 snRNA. SNRNP70, together with SmD3, coordinates SNRPC to support the base-pairing interaction between the 5′-end of the U1 snRNA and 5′ss on the pre-mRNA substrate through its zinc-finger domain [65,69,92]. The recruitment appears to be mediated by RNA polymerase II while it is synthesizing the pre-mRNA; and dependent on the presence of members of the SR family of RNA splicing enhancer proteins [93]. In yeast, recognition of the 5′ss was shown to be supported by U1C/SNRPC, Luc7/LUC7L2, Nam8/TIA-1 [94] and Prp39/PRPF39 [95]. SF1 binds to the BPS [94] and U2 snRNP auxiliary factor 65 kDa subunit (U2AF65) and 35 kDa subunit (U2AF35) are recruited to the PPT and intronic 3′ ss, respectively, of the target pre-mRNA [96,97]. Subsequently, on short introns U2 snRNP is recruited through interacting with U1 snRNP and SF1, replacing SF1 at the BPS. The association of U2 snRNP is further stabilized by U2AF65 [73]. On long introns, U2 snRNP is also recruited to SF1 and U2AF65 near the 3′ss and associates with U1 snRNP, snRNP but positions the U1 snRNP to the downstream 5′ss of the next intron [91]. ATP-dependent RNA helicase DDX46 is required for the transition from complex E to the pre-spliceosome A, and facilitates conformational changes within the U2 and the interaction between the U2 and U1 snRNPs [71]. DDX46 remodels the U2 snRNA, allowing its BSL to bind to the BPS in the intron in an ATP-dependent manner, where the adenosine in the YUNAY consensus sequence is excluded, which is important for later catalysis in the splicing reaction. In yeast, Prp39 anchors U2 snRNP to U1 snRNP by acting as a bridge between the U1C protein and U2 small nuclear ribonucleoprotein A′ (U2A′). In humans, an interaction between PRPF39 and SNRPC is also observed, but is not crucial for complex A formation [98]. This is in line with the exon definition model, where the recruited U1 snRNP and U2 snRNP are to participate in splicing of different introns on either side of the exon. For splicing of transcripts with long introns, neighboring exons must be juxtaposed, existing U1 snRNP-U2 snRNP interactions across exons need to be broken; and new contacts spanning introns need to be established. This transition is still poorly understood, but the process is inhibited by hnRNPI. In the presence of hnRNPI, spliceosome assembly with U1 and U2 snRNPs recruited around exons stalls in an A-like complex [99], showing that the transition occurs prior to U4/U6.U5 tri-snRNP recruitment. Recently, a model for early spliceosome assembly was proposed that unifies the intron definition and exon definition models [94]. Based on cryo-EM analysis of in vitro assembled complexes E and A it was concluded that the same structure can be formed across either an intron or an exon. Structural constraints of complexes formed across short exons make it difficult for the U4/U6.U5 tri-snRNP to subsequently join the spliceosome. This is postulated to be a main trigger for remodeling U1 snRNP-U2 snRNP interactions into an intron-spanning complex, allowing further spliceosome assembly [94].

The pre-activated spliceosome or complex B is formed when the U4/U6.U5 tri-snRNP is recruited. The U5-specific PRPF8 with its N-terminus is able to interact with EFTUD2, DDX23 and the U5 snRNA, which interacts with the pre-mRNA substrate [100]. As was shown in yeast, Prp8/PRPF8′s C-terminus interacts in U5 with the N-terminal helicase domain of SNRNP200. SNRNP200′s C-terminus interacts with EFTUD2 and Ubiquitin Specific Peptidase 39 (USP39). Positioned at the interface of U4/U6 and U5 snRNPs, USP39 is crucial for the stability of the tri-snRNP [101]. Moreover, USP39 is postulated to keep the SNRNP200 RNA helicase positioned away from the U4/U6 duplex, preventing premature unwinding of the U4/U6 snRNAs and thereby of spliceosome catalytic activity in the pre-catalytic stage [78].

During the association of the tri-snRNP with the U2 snRNP, the U1 snRNP places its snRNA between the U4 snRNA and PRPF8, while the U1 SmE and SmG interact with U5-specific ATP-dependent RNA helicase DDX23 [100]. DDX23 unwinds the U1 snRNA:5′ss duplex, and is therefore required for B complex formation, as U6 snRNP replaces U1 snRNP at the 5′ss. Mutations in the DDX23 domain involved in ATP hydrolysis stall the spliceosome before complex B formation, in which U1 snRNP remains associated with the pre-mRNA and the tri-snRNP is not stably integrated yet [102]. The U4 and U6 snRNAs partly form a duplex within the tri-snRNP, rendering U6 snRNA in its inactive configuration. EFTUD2 is also involved in the recruitment of the NTC and NineTeen/Prp19 complex related (NTR) complexes. Recruitment of the NTC and NTR induce conformational changes within the snRNPs necessary for the formation of the active site for the splicing reaction, such as basepairing of the ACAGAGA box of the U6 snRNA with the 5′ss [103,104,105]. As was demonstrated in yeast, pairing of U6 snRNA with the 5′ss occurs prior to U4:U6 duplex unwinding within the embrace of PRPF8 and represents a checkpoint for proper complex B assembly [105].

#### 2.1.5. Dynamic Composition of the Spliceosome: Activation and Catalytic Steps

The unwinding of the U4/U6 snRNAs represents a checkpoint for complex B activation, creating complex Bact. To achieve this, USP39 dissociates, which repositions SNRNP200 and induces conformational changes in SNRNP200 that prompt its helicase activity. During activation of catalytic activity, additional conformational changes occur in the tri-snRNP complex. DDX23 migrates from the outer side of the tri-snRNP towards the RNase H domain of PRPF8 in the center of the complex where the 5′ss basepairing is switched from U1 to U6 snRNA [78]. This results in the release of the U1 snRNP, thereby preventing steric clash of this snRNP with SNRNP200 [98]. This transition from U1 to U6 5′ss basepairing is further supported by the U4/U6.U5 tri-snRNP specific SNRNP27, as was demonstrated in *C. elegans* [106]. Upon the 5′ss transition to the U6 snRNA, SNRNP200 unwinds the U4:U6 snRNA duplex resulting in the dissociation of U4 snRNP from the spliceosome. This allows the 3′ end of the U6 snRNA to basepair with the 5′ end of the U2 snRNA (forming helix I); and also to form a highly conserved internal stem loop (ISL) within the U6 snRNA [107]. Meanwhile, PRPF8 undergoes rearrangements from an open to a closed conformation, as a pocket must be formed to harbor the newly formed U2:U6 duplex and the U5 snRNA SL1, which is necessary to form the active catalytic site [78]. Both the helix I and ISL are involved in the coordination of catalytic metal ions. Overall, SNRNP200, EFTUD2 and PRPF8 are essential for the transition from the pre-catalytic B complex to the activated B^act^ complex. In this intermediate, the active site is cradled by PRPF8, consisting of helix I of the U2:U6 duplex, ISL of U6 snRNA, five Mg^2+^ ions and SL1 of the U5 snRNA [4]. The SL1 of U5 snRNA is basepaired with the 5′ exon [77]. Moreover, within the active site, a triplex structure is formed by several nucleotides of the U6 snRNA [77,108,109].

ATP-dependent RNA helicase-like protein Prp2/DHX16 promotes the transition from the activated B^act^ complex to the catalytically active B* complex [110], through rearrangement of the U2 snRNP around the U2 snRNA:BPS duplex [77,109]. Moreover, several nucleotides of the U6 snRNA are involved in the coordination of Mg^2+^ metal ions via binding to their phosphate groups [77,111]. Of these, two are directly involved in catalysis of the splicing reaction, and the other three fulfill more structural roles [4]. The rearrangements around the U2 snRNA:BPS are supported by step I factors YJU2 [4,109,111,112,113] and CWC25 [4,113] and the presence of one of the two catalytic metal ions (M2) activates the 2′-OH BPS adenosine to perform step I of the splicing reaction; a nucleophilic attack on the phosphorous atom of the 5′ss G nucleotide in which the covalent bond between the 5′ exon and 5′ss is broken. A phosphodiester bond is formed between the BPS adenosine and the guanine of the 5′ss, resulting in an intron-3′exon lariat structure and a free 5′ exon, which remains anchored to loop I of U5 snRNA [4,114] and is stabilized by Prp8 [112]. This represents the complex C spliceosome intermediate [114,115].

Transition from step I complex C into the step II catalytically activated C* complex is facilitated by ATP-dependent RNA helicase DHX38, which triggers the release of step I factors [116] and a conformational change in the Prp8 -encapsulated active site, leading to the replacement of the lariat by the 3′ss at the active site [112]. The introduction of the 3′ss in the active site is stabilized by SLU7 [113,116]. During the second transesterification reaction, supported by Prp8/PRPF8, Prp17 and Prp18, the 3′-OH of the 5′ exon performs a nucleophilic attack on the phosphate of the 3′ exon [114]. This results in a covalent bond between the two exons and an intron lariat still bound by spliceosomal components: the post-splicing complex P.

The exon junction complex (EJC) is formed over the ligated exons and connects splicing to other downstream mRNA processes, such as export, translation and NMD [117,118]. The ligated exons are bound by U5 snRNA loop I and the 3′ end of the ligated exon pulled from the intron lariat and the spliceosome by the ATP-dependent RNA helicase Prp22/DHX8 while the intron lariat is released by the ATP-dependent RNA helicase Prp43/DHX15, giving rise to the intron lariat spliceosome intermediate and the spliced mRNA [3]. In a final step, Prp43/DHX15 releases the U2, U5 and U6 snRNPs and the NTC and NRC from the intron lariat, facilitating the recycling of these spliceosome components into the next splicing reaction.

### 2.2. Regulation of (Alternative) mRNA Splicing

Although for most human genes that undergo alternative splicing one mature transcript variant is usually dominant, representing at least 30% of the total transcripts, they express many splice variants simultaneously [119]. Exons that are included in all transcript variants are called constitutive exons; those that are present in only a subset of the transcripts cassette exons or alternative exons. The exclusion of these cassette exons represents one example of the most common form of AS—namely exon skipping. Other forms of AS are intron retention, selective incorporation of mutually exclusive exons and the usage of alternative 3′ or 5′ splice sites, resulting in exclusion of part of an exon or inclusion of part of an intron. The different types of AS events are illustrated in Figure 4a.

AS is regulated through many different factors and intrinsic properties of the pre-mRNA sequence, reviewed extensively elsewhere [120,121,122] and illustrated in Figure 4b. Foremost, the splice sites themselves are involved in mRNA splicing regulation. They have limited sequence constraints. According to their compliance to the consensus splice site sequence, a subdivision can be made in “strong” and “weak” splice sites. However, the strength of a splice site is not only dependent on its sequence, but is also influenced by the gene context. This is illustrated in a study in which libraries of BRCA2, SMN1 and ELP1 minigenes harboring sets of randomized 5′ss were transfected into cells and their splicing products were analyzed by RT-PCR. The authors found that the randomized 5′ss brought about very similar splicing patterns (as demonstrated by the percent spliced-in values) within the same gene, but very different splicing patterns in the three genes [15].

The strength of the splice site determines the efficiency of recognition by the snRNPs and other splice factors and competition between splice sites leads to AS. Strong splice sites usually delineate constitutive exons. Splice site usage is influenced by cis-acting RNA sequence elements, including intronic splicing enhancers (ISEs), intronic splicing silencers (ISSs), exonic splicing enhancers (ESEs) and exonic splicing silencers (ESSs). These cis-acting regions can be bound by trans-acting regulatory proteins. The most important trans-acting factors in mRNA splicing are heterogeneous nuclear ribonucleoproteins (hnRNPs) and serine and arginine-rich (SR) proteins. SR proteins are generally regarded as splicing enhancers and predominantly bind to ESEs, whereas hnRNPs appear to inhibit splicing, through preventing assembly of the spliceosome at splice sites. Moreover, SR proteins indirectly promote splicing by impeding hnRNP-mediated repressive effects. The main cis-acting elements and trans-acting proteins are depicted in Figure 4b. However, opposing roles for these proteins have also been described and seem to depend on the sequence of the cis-acting element, on the position with regards to the target site in the pre-mRNA [123], and whether the surrounding exons are constitutive or not [124,125].

Obviously, mutations in the splice site sequences, which are associated with disease but rare in lung cancer (see Section 3), affect their recognition by the mRNA splicing machinery and thus AS. In addition, mutations elsewhere on the pre-mRNA transcript may introduce cryptic splice sites that compete with the canonical sites. Here, in particular the role of Alu retrotransposons is worth mentioning. These most abundant transposable elements in the human genome are present in most primary transcripts [126]. Alu elements contain cryptic splice sites and when they are inserted in an intron in the antisense orientation, their poly(A) tract can be recognized as PPT sequence, promoting recruitment of the spliceosome to the cryptic splice site [126]. If multiple Alu elements are integrated in a long intron they can together delineate a cryptic exon. When these cryptic sites are used by the mRNA splicing machinery, this creates a new exon. This event known as exonization contributes to evolutionary complexity. In addition, a systematic analysis of Alu elements integrated in introns near exons with rather weak splice sites showed that they can alter exon incorporation efficiencies [127].

#### 2.2.1. Trans-Acting mRNA Splicing Factors

hnRNPs inhibit mRNA splicing through interfering with the core spliceosome. These factors accomplish this by binding to ISSs or ESSs and are thought to sterically hinder the interaction of the pre-mRNA with core splice factors. Inhibition of the spliceosomal machinery typically leads to exon exclusion. There are 32 hnRNP proteins known to date (according to KEGG, amiGO and Reactome database) that can bind distinct sequences in the pre-mRNA and were shown to play a role in the splicing of different genes. For example, hnRNPL recognizes CA-rich RNA sequences. For the CD44 gene, splicing of the variant exon 15 (or V10) was enhanced or abolished by the removal or addition of these repeats, respectively. Inhibition of splicing was explained through inhibition by hnRNPL of U2AF65 binding to the PPT [128]. Another example is hnRNPC, which was found to compete with U2AF35 for binding to uridine-rich 3′ss. This prevents aberrant inclusion of cryptic exons, amongst others for the apoptotic regulator BAX mRNA [97]. Interestingly, hnRNPC was also shown to protect the transcriptome against exonization by competing with U2AF65 for binding at cryptic splice sites created by integrated Alu elements [97]. Another RBP, hnRNPI, was speculated to act in a similar manner as hnRNPC [97]. The hnRNP-related Poly(C)-Binding Proteins (PCBPs) promote inclusion of cassette exons by binding to cytosine-rich PPTs upstream of exons [129]. Interestingly, while hnRNPs are primarily mRNA splicing inhibitors, hnRNPH can either promote or repress mRNA splicing when binding to different elements on pre-mRNAs. hnRNPH binds to poly-guanine sequences containing GGG triplets (so-called G-runs). Intronic G-runs nearby 5′ splice sites function as ISE motif, whereas G-runs in exons are commonly ESS motifs. Hence, recruitment of hnRNPH to a G-run ISE motif increases the strength of the neighboring 5′splice site thus promoting splicing [130], whereas hnRNPH inhibits splicing when it binds to a G-run ESS [131].

The SR protein family consists of 19 members (according to KEGG, amiGO and Reactome databases) and these proteins contain one or two N-terminal RNA recognition motifs (RRMs) and one arginine-serine (RS) domain consisting of at least 50 amino acids with successive RS or SR dipeptides at the C-terminus. The RRM binds certain 4–8 nucleotide consensus sequences in the target pre-mRNA while the RS domain is responsible for protein–protein interactions. With other proteins in the spliceosome. For instance, SRSF1 recognizes 5′ss through its RRM domain [132] and directly interacts with SNRNP70, recruiting the U1 snRNP to the 5′ splice site [133], where the U1 snRNP further stabilizes the interaction between SRSF1 and the pre-mRNA [132]. SR proteins bind to pre-mRNA with low affinity and specificity, which contributes to the highly dynamic nature of the spliceosome [134]. SR proteins are subject to post-translational modifications, such as phosphorylation by SR protein kinases (SRPKs) and CDC-like kinases (CLKs); and dephosphorylation by serine/threonine-protein phosphatases. Hypophosphorylation of SRSF1 results in intramolecular interactions within SRSF1 that prevent its binding to SNRNP70, decreasing the recruitment of U1 snRNP to the pre-mRNA and thereby reducing mRNA splicing. Conversely, hyperphosphorylation of SRSF1 results in the formation of a splicing-promoting complex consisting of the ESE, SRSF1 and U1 snRNP [133]. Interestingly, upon phosphorylation by CLK1, SRSF1 was found to remain bound to CLK1. SRSF1 was released from CLK1 upon SRPK1 interacting with CLK1, thereby becoming active and able to recruit U1 snRNP to the pre-mRNA. Thus, SRPK1 and CLK1 activities were both needed to allow mRNA splicing [135].

Apart from SR and hnRNP proteins, several other proteins in the spliceosome were also shown to affect AS. For example, the complex B-specific proteins SMU1 and protein RED are involved in the splicing of short introns (i.e., introns in which the distance between the 5′ss and the BPS is relatively short). Silencing of these proteins resulted in an increased retention of this type of introns, but also in skipping of cassette exons and alternative 5′ and 3′ ss selection [136]. In another study, it was found that knockdown of SMU1 and protein RED show overlapping effects on AS [137]. For another complex B-specific protein, microfibrillar-associated protein 1, knockdown led to similar effects on intron splicing as knocking down SMU1 and protein RED, but to a different subset of introns [136]; and the KHDRBS1 protein was found to interact with U1-A protein, thereby promoting the recruitment of the U1 snRNP at 5′ss [138]. Nuclear cyclophilins are additional examples of proteins other than SR or hnRNP proteins that can regulate RNA splicing, specifically through interfering with spliceosome assembly [139]. In addition to these mRNA splicing enhancers and inhibitors that are widely expressed in many tissues, there are also highly tissue-specific trans-acting factors that are expressed almost exclusively in e.g., neurons or muscle cells. For the purpose of this review, these mRNA splicing factors are disregarded.

#### 2.2.2. Effect of Secondary mRNA Structure

The secondary structure of the pre-mRNA substrate can also affect (alternative) splicing (reviewed in [140]). For example, stem-loop hairpin structures caused by intramolecular basepairing alter the local accessibility for trans-acting proteins. If a cis-acting enhancer or silencer motif is present in the stem of a hairpin structure, it is generally inaccessible for protein binding and thus dysfunctional for mRNA splicing regulation. Conversely, ESEs that are located immediately downstream of a hairpin structure usually exhibit strong enhancer activity. Hence, mutations in the pre-mRNA sequence that change its secondary structure by increasing or decreasing the stability of hairpins may affect splice factor binding and thus AS. In addition, mutations near splice sites may create hairpins that sequester the splice site sequence, thereby inhibiting recruitment of the spliceosome. In contrast, certain RBPs bind specifically to hairpins. For example, MBNL1 was shown to bind a hairpin sequence that contains a binding site for U2AF65 in its loop portion. MBNL1 and U2AF65 compete for binding, where MBNL1 inhibits U2AF65 binding and thus U2 snRNP recruitment when the intron adopts a hairpin structure, whereas U2AF65 binds to allow splicing when the sequence is in its single-strand fashion [141]. Another secondary RNA structure that affects splicing efficiency is the so-called G-quadruplex structure. This can be formed by a conserved sequence motif comprising at least four tracts of GG dinucleotides, folding into a helix consisting of stacked planar structures that are held together through Hoogsteen hydrogen bonding. In pull-down assays with G-quadruplex-forming RNA oligonucleotides the spliceosome proteins U2AF65, SRSF1, SRSF9, hnRNPF, hnRNPH and hnRNPU were identified [142]. The effects of G-quadruplex sequences on AS probably depend on their position within the RNA sequence and on which trans-acting protein they bind. At least for hnRNPF and hnRNPH there is suggestive evidence that their binding to G-quadruplex structures changes mRNA splicing [143,144]. Finally, RNA duplex structures formed by intramolecular interaction between sequence motifs located at sometimes very large distance in introns were found to determine AS. An example of this is the alternative exon incorporation in the FGFR2 gene causing different isoforms expressed in different cell types. The formation of the duplex structure was concluded to function solely to juxtaposition otherwise distant cis-acting elements [145].

#### 2.2.3. Effect of mRNA Elongation Rate

As mRNA splicing occurs co-transcriptionally, the rate by which RNA polymerase II elongates the pre-mRNA also influences the mRNA splicing process (Figure 4b). Intuitively, one could argue that if transcription rate is high, producing a high concentration of pre-mRNA substrate for the splicing reaction, components of the mRNA splicing machinery could become limiting, resulting in less efficient splicing. Indeed, studies in yeast revealed that pre-mRNAs compete for the limited supply of splicing machinery components [146]. However, Ding and Elowitz observed the opposite in mammalian cells [147]. By measuring constitutive mRNA splicing efficiencies at transcription active sites in individual cells, they found that splicing efficiency increased with increasing levels of transcription. Since the efficiency of splicing regulates the nuclear export of mature mRNAs, they hypothesized that this effect amplifies the expression of more strongly transcribed genes and reduces expression of low-level transcribed genes, thus contributing to gene expression modulation. Conversely, mRNA splicing also influences RNA elongation. Alexander et al. observed in yeast that RNA polymerase II paused periodically around the 3′ end of introns [148]. This coincided with—and was dependent on—the recruitment of U2 and U5 snRNPs. RNA polymerase II that stalled near the 3′splice site was hyperphosphorylated in its carboxy-terminal domain known to recruit RNA processing factors. The authors propose that the mRNA splicing machinery controls a transcription checkpoint that is activated during formation of the step II catalytically activated spliceosome and released upon completion of the second step of the RNA splicing reaction. Although there are many candidates, the RNA splicing factor responsible for activating the proposed transcription checkpoint and its possible association with RNA polymerase II has not yet been ascertained.

The RNA elongation rate might also affect the fidelity of the splicing reaction and splice site preference. Regulation of transcription initiation and elongation is reviewed elsewhere [149]. Critical factors controlling productive elongation by RNA polymerase II include nucleosome positioning and histone modification. These epigenetic processes influence the speed at which RNA polymerase II can move forward along the DNA template. How this may affect mRNA splicing is already extensively discussed by others [150]. During transcription, for any set of alternative splice sites, the upstream located splice site is created sooner and thus available for recognition by the RNA splicing machinery earlier, giving it a selection advantage over more downstream splice sites. Thus, in theory, the slower the elongation rate, the longer the time difference in creation of the competing splice sites; and thus the larger the preference for upstream splice site selection. However, the kinetic coupling of pre-mRNA elongation and splicing processes appears more complicated. A genome-wide analysis of AS at different elongation rates revealed that, while the elongation rate clearly affected both constitutive and alternative mRNA splicing, elongation-rate dependent AS events were not always consistent with this reasoning [151]. Exons that were included more frequently at slow elongation rates usually had weaker splice sites than those that were included more often at high elongation rates. However, when the elongation rate was experimentally increased or reduced, AS of many genes was affected in the same way, rather than having opposing effects, suggesting that a proper balance of RNA splice variants requires an optimal RNA transcription rate [151]. It also suggests that mRNA splicing patterns could change if the transcription rate is changed in response to physiological stimuli or disease processes. This, obviously, is relevant in cancer, where transcription elongation rates of many genes are altered by oncogene expression [152,153].

#### 2.2.4. Effect of Chromatin Structure

Apart from epigenetic processes affecting mRNA splicing through the kinetic coupling of transcription and splicing discussed above, there is also evidence that chromatin organization and histone modifications have more direct effects on mRNA splicing, by contributing to exon definition and splice site choice (reviewed in [154,155]). Nucleosomes are found enriched at exon-intron junctions suggesting that they play a role in exon definition [156,157]. Moreover, in alternatively spliced genes they are more highly enriched around included exons than around excluded ones. Together, this strongly argues for a function of nucleosome positioning in regulating splicing [156]. Also, certain histone modification marks are found enriched at exons, more than would be expected as a consequence of nucleosome distribution [157]. For some histone modifications a mechanism by which they modulate mRNA splicing was revealed. For example, H3K4me3 was shown to bind the U2 snRNA via the chromatin remodeling protein CHD1; and this promoted recruitment of the U2 snRNP to the pre-mRNA branch site [158]. Another example is the recruitment of hnRNPI via MRG15 binding to H3K36me3 [159]. Thus, histone modifications appear to influence AS by recruiting mRNA splicing factors to the pre-mRNA substrate via chromatin-binding adapter proteins.

### 2.3. Other RNA Processing Activities in the Supraspliceosome

In addition to the mRNA splicing machinery, the supraspliceosome harbors associated enzymes with 5′-RNA end capping, 3′-RNA end cleavage, RNA polyadenylation and RNA base editing activities (Figure 5). The co-transcriptional conjunction of these processes in close proximity in the supraspliceosome makes many of these processes interdependent. In this section, we discuss these processes and their effects on mRNA splicing.

#### 2.3.1. mRNA 5′ Capping

RNA 5′ capping (Figure 5a) involves the attachment of an N7-methylated guanosine (m7G) to the first nucleotide of the pre-mRNA, which protects it from 5′ to 3′ exonuclease cleavage, supports the initiation of translation and recruits the RNA splicing machinery as well as the factors required for 3′ polyadenylation and nuclear export. The 5′ capping takes place in three steps, in humans catalyzed by RNA Guanylyltransferase and 5′-Phosphatase (RNGTT) and RNA guanine N7-methyltransferase (RNMT) (reviewed by [160]). Additionally, the +1 nucleotide can be methylated at the 2′O of its ribose by mRNA Cap 2′-O-methyltransferase. The cap structure is bound by the cap binding complex (CBC) in the nucleus. The CBC proteins CBP20 and CBP80 were found in the supraspliceosome [10]. The CBC recruits spliceosomes and factors involved in 3′ end processing, RNA export and NMD to the mRNA. After guiding the transport of the processed mRNA into the cytoplasm, the CBC is replaced by the eukaryotic translation initiation factor 4E (eIF4E) to start mRNA translation [160].

#### 2.3.2. mRNA 3′ End Cleavage and Polyadenylation

The mechanism and structural details of the 3′RNA end cleavage and polyadenylation machinery that adds a 3′ polyadenosine (poly(A)) tail to most mRNAs to safeguard mRNA stability and regulate translation are reviewed by Kumar et al. [161] and is illustrated in Figure 5b. Co-transcriptionally, the AAUAAA hexanucleotide within the polyadenylation signal (PAS) on the pre-mRNA is recognized by a protein complex composed of the cleavage and polyadenylation specificity factor 1 (CPSF1), CPSF4, WD repeat domain 33 (WDR33) and pre-mRNA 3′-end-processing factor FIP1. The CPSF proteins associate with the carboxy-terminal domain of RNA polymerase II to regulate their activities on the mRNA. This leads to the cleavage of the pre-mRNA, 10–30 nucleotides downstream of the PAS, by CPSF3. Finally, Poly(A) polymerase (PAP) interacts with FIP1 and adds the poly(A) tail to the cleaved (pre-)mRNA. 3′-end processing complexes involved in the cleavage of the pre-mRNA downstream of the PAS, i.e., NUDT21, CPSF6, CPSF7 [24,162], CPSF1, 2, 3, 4 and FIP1 [24,162], CSTF 1, 2 and 3 and symplekin [10,24,162] and polyadenylate-binding protein 2 [24,162] were all shown to be present in the supraspliceosome. Most of the mRNA splicing reactions on the pre-mRNA are completed before 3′RNA end cleavage and polyadenylation occur, but this may be completed for some introns on the polyadenylated RNA [12]. Co-transcriptionally, U1 snRNP not only plays a crucial role in the mRNA splicing reaction by defining the 5′ss of introns, but also protects the pre-mRNA from premature cleavage and polyadenylation at cryptic PASs in introns; a process coined telescripting [163,164]. Telescripting requires specific base pairing of the U1 snRNA to the pre-mRNA. Presumably, the binding of U1 snRNP to the intron physically competes with binding of the cleavage/polyadenylation machinery at nearby sites. Since 5′ splice sites could be at too large a distance of cryptic PASs in long introns, cryptic 5′ splice sites may primarily serve to protect the pre-mRNA from premature termination by recruiting U1 snRNP [164]. Modulation of PAS suppression by U1 snRNP also affected alternative splicing patterns [163].

#### 2.3.3. mRNA Internal Adenosine Methylation

The most prevalent internal mRNA modification in eukaryotes is methylation of adenosine residues at their N6-position (depicted as m^6^A) in certain consensus sequence motifs that are enriched near stop codons and in long exons [165] (Figure 5c). The m^6^A modification is important for gene expression regulation and is reversible. It is installed co-transcriptionally on the mRNA by methyltransferases, most notably the nuclear complex consisting of METTL3/METTL14 heterodimers with accessory proteins; and can be removed by the demethylases FTO and ALKBH5. Dysregulated expression of these m^6^A modification enzymes as well as of proteins that bind to m^6^A sites is associated with cancer [166]. While many effects of m^6^A modification on gene expression involve nuclear export and cytosolic processes such as mRNA stability and translation efficiency, which for the purpose of this review are disregarded, m^6^A modification also affects mRNA splicing. The modification changes the mRNA structure, increasing the accessibility for binding of mRNA splicing silencers hnRNPC and hnRNPG [167,168]. Consequently, knockdown of METTL3 or METTL14, which reduces m^6^A modification, induced AS with similar patterns as knockdown of hnRNPs. Conversely, depletion of demethylases ALKBH5 or FTO increased m^6^A levels and changed mRNA splicing patterns [169,170]. ALKBH5 expression was shown to promote nuclear localization of SRPK1 and this was associated with increased levels of phosphorylated SRSF1 [170], which activates this SR protein to recruit U1 snRNP [135]. Depletion of FTO promoted recruitment of SRSF2 and increased exon inclusion, presumably by increasing methylation of adenosines in ESEs [169]. Furthermore, the nuclear m^6^A-binding protein YTDHC1 was shown to bind several members of the SR family of proteins [171]. When YTDHC1 bound to an m^6^A motif in an exon, this promoted binding of SRSF3 to the mRNA and exon inclusion. At the same time, YTDHC1 inhibited binding of SRSF10, which would otherwise promote exon skipping. Hence, m^6^A modification clearly favored inclusion of the exon. Together, these observations suggest that structural changes in the pre-mRNA imposed by methylation contribute to creating cis-acting signals for regulating mRNA splicing.

#### 2.3.4. mRNA Base Editing

RNA editing consists of adenosine-to-inosine (A-to-I) substitution, cytosine-to-uridine (C-to-U) substitution, U insertion/deletion, C or guanosine (G) insertion, 2′-O-methyl ribose nucleotide modification, and U-to-pseudo U conversion, all resulting in an altered RNA sequence compared to the genetic code on the DNA template [172]. Of these, A-to-I and C-to-U editing (Figure 5d) are most thoroughly described. A-to-I editing is the most prevalent form of RNA editing in mammalian cells. The reaction is catalyzed by C6-adenosine deaminases acting on RNA (ADARs). Three mammalian ADARs are known to date (ADAR1-3), of which ADAR1 and 2 possess catalytic activity, propagated through their C-terminal deaminase catalytic domains. Site-specificity for editing in exons is dictated by cis-acting complementary repeats in the intronic sequence (the editing site complementary sequence; ECS) adjacent to the exon containing the adenosine that is to be edited. An imperfect duplex structure is formed between the intronic sequence and spanning the exonic editing site, which is essential for recognition by ADARs through their double-stranded RNA (dsRNA) binding motifs [173]. C-to-U editing is performed by the cytidine deaminase apolipoprotein B mRNA editing enzyme (APOBEC) family. APOBEC1 binds to single-stranded RNA through its cytidine deaminase domain. Editing activity of APOBEC1 is dependent on a cis-acting 11 nucleotide sequence adjacent to the editing site and guidance by RNA-binding protein cofactor A1CF towards the editing site [174]. At least the RNA editing enzymes ADAR1 and ADAR2 were found in the supraspliceosome, in association with Sm and SR proteins present in the spliceosomes, while retaining their A-to-I editing activity [11]. RNA base editing can influence simultaneous mRNA splicing and vice versa. While most A-to-I editing sites are situated in non-coding regions of the mRNA, such as intronic short Alu repeats and 3′-untranslated regions (3′-UTRs) [175], editing may change splicing factor recognition sites or cis-acting elements on the template mRNA, thereby changing the efficiency of splice site recognition and thus causing AS. The mRNA splicing machinery recognizes an inosine as guanine. Thus, A-to-I editing at an intronic AA dinucleotide can create an AI sequence that effectively mimics the conserved AG sequence found at 3′ splice sites [176] creating an alternative splice acceptor site, or, conversely, the strength of a genuine 3′ss can be dramatically reduced if it is converted from AG to IG [177]. On the other hand, mRNA splicing may remove the intronic ECS required to form the dsRNA editing template and thus inhibit RNA editing in an exon. In addition, mRNA splicing factors and base editor enzymes in the supraspliceosome could compete for binding to the mRNA template through steric hindrance. For example, it has been observed that binding of SRSF9 to a mouse minigene containing the ECS of the voltage-gated calcium channel CaV1.3 mRNA inhibited A-to-I editing by ADAR2 [178]. Conversely, binding of ADAR2 to a dsRNA template formed by intron 2 and exon 3 sequences of the RELL2 gene blocked access of splicing factor U2AF35 to the 3′ splice site, thereby reducing exon 3 inclusion [179]. This exon skipping event led to degradation of RELL2 transcripts through NMD, possibly contributing to tumorigenesis [179]. RNA splicing can also influence RNA editing via AS of ADAR transcripts. For instance, ADAR2 pre-mRNA carries a cassette exon in intron 7 that contains multiple stop codons through which transcripts with this exon are targeted for NMD. In tissues in which inclusion of the cassette exon is high, ADAR2 expression is decreased and ADAR2-mediated A-to-I editing is low [180].

#### 2.3.5. Intronic Pri-miRNAs

Intronic pri-miRNAs and key microprocessor proteins Drosha and DGCR8 were found associated with the supraspliceosome [181]. This suggested that intronic pri-miRNAs are processed in the supraspliceosome. Indeed, for the miR-106b-25 cluster that is located in intron 13 of the MCM7 pre-mRNA and harbors miRNAs 106b, 93 and 25, processing from pri-miRNA to pre-miRNA in the supraspliceosome was demonstrated [181]. Interestingly, intronic miRNA processing and mRNA splicing appeared interdependent. Usage of an alternative 3′ splice acceptor site in MCM7 intron 13 added the miR-25 sequences to an extended exon 14, rather than splicing this section out and subjecting it to the microprocessor complex. Hence, this AS event resulted in lower miR-25 levels. Specific inhibition of this AS event using antisense morpholinos, favoring inclusion of the miR-25 pre-miRNA hairpin sequences in the intron, increased miR-25 expression. General inhibition of mRNA splicing using the SF3b1 inhibitor spliceostatin A reduced alternative splice site usage more than constitutive splice site usage, thereby also increasing miRNA expression. Conversely, inhibition of miRNA processing through silencing of Drosha increased AS, possibly because the alternative splice site near the hairpin in the intron is more accessible to the splicing machinery in the absence of Drosha [181]. Together, these observations support the notion that intronic pri-miRNAs are processed in the supraspliceosome in conjunction with the mRNA splicing process before being handed over to the nuclear export machinery. A comprehensive analysis of miRNAs in the supraspliceosome revealed that while most were intronic miRNAs, a substantial fraction was of intergenic origin [182]. In addition, the majority were mature miRNAs that are apparently shuttled from the cytoplasm to the nucleus; and most of these were not detected in precursor format [182]. Many miRNAs were found enriched in the supraspliceosome, suggesting selective association with the supraspliceosome. Hence, their presence in the supraspliceosome is not a simple reflection of their production in the supraspliceosome during the mRNA splicing process, but suggests an undiscovered function for miRNAs in this complex. Compelling evidence was obtained to suggest that at least some of these miRNAs regulate gene expression and perhaps splicing [182].

#### 2.3.6. mRNA Transport

Aside from proteins with direct roles in RNA processing and editing, factors involved in mRNA transport (TRanscription-EXport (TREX) complex) and surveillance and localization (EJC) were found to be associated with the supraspliceosome [162]. Interestingly, also nuclear matrix and filament proteins were detected, as well as proteins of the nuclear pore complex. It was speculated that these proteins might play a role in guiding the mRNA from the splicing machinery to the nuclear pores.

Recently, more striking examples of the interaction between mRNA splicing and other mRNA processing steps were described. U1 snRNP was found to be involved in the nuclear retention of long noncoding RNAs (lncRNAs), by tethering these molecules to chromatin. This occurs co-transcriptionally and is dependent on RNA polymerase II. Interestingly, inhibition of U2 snRNP through treatment with the splicing inhibitor E7107 displayed similar, albeit more subtle effects on chromatin retention of lncRNAs as compared to interfering with U1 snRNP. In contrast, inhibition of the U4/U6.U5 tri-snRNP by the drug isoginkgetin did not have any effect [183]. In another study, U1 proteins SNRNP70 and SNRPA, but also SmD2 were shown to be involved in nuclear retention of spliced lncRNAs [184]. Hence, U1 appears to display multiple functions to ensure transcriptome integrity distinct from mRNA splicing. For the SF3b complex, a role in mRNA export was also described. This is, however, independent of U2 snRNP and is achieved through interacting with the TREX complex [185]. Finally, the U4/U6.U5 tri-snRNP has been implicated in maintaining proper chromatin cohesion and enabling mitotic progression through its interaction with cohesion [186].

Together, the supraspliceosome offers a platform for coordinated mRNA processing, splicing, editing, surveillance and transport. This machine controls all nuclear steps necessary to obtain mature mRNA that can be translated to proteins in the cytoplasm.

## 3. RNA Splicing Dysregulation in Lung Cancer

As discussed above, mRNA splicing is a highly regulated process that is controlled by sequence elements on the pre-mRNA template and many RNA processing proteins in the supraspliceosome. Hence, there are many elements and processes that can become perturbed causing dysregulated mRNA splicing. Genetic instability and high mutation rates in cancer allow for frequent occurrence of such perturbations in cancer [187]. Dysregulated mRNA splicing plays a key role in multiple facets of the disease. Here, we mainly focus on the role of dysregulated mRNA splicing in lung cancer and explore its effects on lung cancer tumorigenesis, disease progression, and response to lung cancer therapy.

### 3.1. Alterations in Cis-Acting Elements on Target Pre-mRNAs

In a genomic analysis of mRNA splicing events in lung cancer cells, Liu et al. [188] identified a number of cancer-specific aberrant splicing events. Only approximately 5% of the differentially spliced genes exhibited splice site variations. Thus, there does not seem to be a prominent role for splice site mutations in lung cancer. Nevertheless, several mutations in splice sites on template pre-mRNAs that contribute to aberrant mRNA splicing events in lung cancer were described. They are shortly discussed in this section and illustrated in Figure 6a.

#### 3.1.1. ARID1A

The first is an AS variant of AT-rich interaction domain-containing protein 1A (ARID1A). ARID1A is a tumor suppressor gene that is frequently mutated across several tumor types [189]. The protein is part of an ATP-dependent chromatin remodeling complex that is required for transcriptional activation of genes that are repressed by chromatin. In non-small cell lung cancer (NSCLC), the incidence of ARID1A mutation is only a few percent [190,191,192]. Nevertheless, loss of ADAR1A expression is associated with worse overall survival among NSCLC patients [190,193]. While many losses of ARID1A expression are the result of other mutations, a small subset of NSCLCs carry splice site mutations [190]. For example, comprehensive analysis of single-nucleotide variants that cause AS on primary cancer specimens revealed an intron retention event in the ARID1A gene of a patient with NSCLC [194] (Figure 6a). A single nucleotide substitution in exon 17 abrogated the 5′ss of intron 17, activating an intronic cryptic splice site in intron 17, resulting in an extended exon 17 that comprises part of intron 17. The resulting mature mRNA is predicted to encode a PTC likely to cause NMD and thus loss of ARID1A protein translation [194]. AS can thus contribute to loss of ARID1A expression in NSCLC, but this seems relatively rare.

#### 3.1.2. TP53

Another AS event caused by splice site mutation in lung cancer is intron retention-induced tumor-suppressor TP53 inactivation. A pan-cancer study focusing on comprehensive identification of mutated splice sites in matched whole-exome and transcriptome sequencing data from 8976 samples across 31 cancer types showed that TP53 was the most frequently altered gene in most cancers, including lung squamous cell carcinomas and lung adenocarcinomas [195]. Among a variety of recurrent splice site mutations in several introns of the TP53 gene, mutational hotspots were identified in the 5′ and 3′ splice sites of intron 4 (Figure 6a). The splice site mutations generated intron retention and exon skipping events. Most of these produced frame shifts that are predicted to result in mRNA degradation through NMD and thus loss of p53 activity.

#### 3.1.3. METΔ14

In a molecular subgroup of NSCLC with poor prognosis, representing approximately 4% of lung adenocarcinomas, somatic mutations in the mesenchymal epithelial transition (MET) gene cause exon 14 skipping [196,197]. The METΔ14 mutation promotes cell proliferation, survival, and metastasis via several pathways. This makes it highly likely that METΔ14 is a cancer driver mutation. MET encodes the c-MET protein, a transmembrane receptor with autonomous phosphorylation activity, belonging to the receptor tyrosine kinase super family. The binding of hepatocyte growth factor to c-MET leads to dimerization of c-MET, tyrosine phosphorylation, and activation of many downstream signaling pathways, such as PI3K-Akt, Ras-MAPK, STAT, and Wnt/ β-catenin. MET exon 14 skipping can be caused by a diversity of deletion mutations in intron 13 spanning the BPS and PPT, some of which extend into exon 14; mutation of the 3′ splice site of intron 13; and a diversity of point and deletion mutations around the 5′ splice site of intron 14 (i.e., in exon 14 as well as in intron 14) [196,198,199] (Figure 6a). The juxtamembrane domain in the c-MET protein section encoded by exon 14 contains a binding site (Tyrosine 1003) for c-CBL E3 ubiquitin ligase. When MET exon 14 skipping occurs, this binding site is missing, resulting in decreased ubiquitination and degradation of c-MET proteins. The reduced c-MET turnover causes sustained c-MET activation and hence promotes proliferation and metastasis of tumor cells [200]. This makes METΔ14 NSCLC particularly amenable to treatment with c-MET targeted tyrosine kinase inhibitors [201,202]. The ATP-competitive c-MET/ALK inhibitor crizotinib [203], followed by highly selective ATP-competitive c-MET inhibitors capmatinib [202] and tepotinib [201] have been successively approved by the United States Food and Drug Administration (US FDA) for the treatment of NSCLC in patients with MET exon 14 skipping alterations. Furthermore, sustained c-MET activation can bypass an inhibited epidermal growth factor receptor (EGFR) pathway via the PI3K-Akt and Ras-MAPK pathways and thus avoid cell killing by EGFR-targeted tyrosine kinase inhibitors (TKIs), promoting cancer cell proliferation, ultimately leading to EGFR-TKI resistance in patients. In clinical practice, acquired EGFR-TKI resistance through MET amplification and METΔ14 mutation can be effectively treated by combination with a c-MET kinase inhibitor [204].

#### 3.1.4. BIM

Apart from splice site mutations, mutations in other cis-acting RNA splicing elements were identified in human lung cancer. For example, an intronic deletion polymorphism of the BIM gene, one of the pro-apoptotic BCL2 family members, has been associated with poor prognosis and intrinsic TKI resistance in EGFR mutant NSCLC cell lines [205]. BIM encodes a BCL2-homology domain 3 (BH3)-containing protein, which promotes cell death via dimerization with other members of the BCL2 family, thereby activating their pro-apoptotic function or inhibiting their pro-survival function, respectively. A common BIM polymorphism was identified in TKI-resistant chronic myeloid leukemia cells from East Asian individuals and confirmed to be present in EGFR mutant NSCLC cells as well. The polymorphism was a deletion of a 2903-bp sequence in intron 2 (Figure 6a). Exons 3 and 4 of BIM are mutually exclusive; they never occur in the same transcript. The deletion favors splicing to exon 3 over exon 4, which generates alternative protein isoforms that lack the crucial BH3 domain needed for dimerization and pro-apoptotic activity and are thus incapable of inducing apoptosis in response to EGFR-TKIs. In individuals with EGFR mutant NSCLC, the BIM intron 2 deletion correlated with a shorter progression free survival [205]. The deleted sequence in intron 2 was found to harbor multiple redundant ISS elements that contribute to skipping of exon 3. The RBPs that regulate exon 3 skipping via binding to these elements have not yet been identified [206].

#### 3.1.5. AIMP2

Another example of a cis-acting regulatory element related to an aberrant splicing event in lung cancer occurs in the Aminoacyl-tRNA synthetases (ARS)-interacting multifunctional protein 2 (AIMP2) gene. Wild type AIMP2 functions as a versatile tumor suppressor factor via inhibition of c-myc expression and Wnt signaling and activation of p53 and TNF-α mediated apoptosis [207,208,209]. In clinical tissues of lung cancer patients, an AIMP2 variant lacking exon 2 (AIMP2-DX2) was found overexpressed in cancerous regions compared to non-malignant regions, whereas expression of the full-length variant was not different. Furthermore, AIMP-DX2 overexpression was associated with lung cancer stage and poorer overall and disease-free survival [210]. Treatment of non-malignant lung cells with a chemical carcinogen increased the expression levels of AIMP2-DX2, suggesting that carcinogenic stress can induce AS of AIMP2. Analysis of genomic DNA from these cells identified several mutations, including an A152G base substitution in an ESE element in exon 2 (Figure 6a). The A152G mutation caused exon 2 skipping by interfering with the interaction between the ESE and SRSF1 [210]. Constitutive overexpression of the AIMP-DX2 variant in transgenic mice increased lung tumor incidence upon chemical induction of carcinogenesis. Conversely, specific silencing of the AIMP-DX2 variant by delivery of siRNA or shRNA to the lungs of mice via inhalation reduced lung tumor growth in transgenic as well as human xenograft models [210]. Hence, the AIMP-DX2 splicing variant appears associated with aggressiveness of lung cancer and a potential target for lung cancer treatment.

### 3.2. Alterations in Trans-Acting mRNA Splicing Factors

The frequent dysregulation of mRNA splicing and infrequent splice site mutation in lung cancer suggest that dysregulation of mRNA splicing in lung cancer is mainly caused by alterations in trans-acting splice factors. These alterations can either be mutations in the splice factor genes or changes in the expression levels or post-translational modification of the splice factor proteins. While mutations in genes encoding spliceosome proteins are commonly associated with hematological malignancies, they are less frequent in solid tumors [211]. The most frequently mutated mRNA splicing factor genes in cancer are SF3B1 and SRSF2. However, in lung cancer, other, generally less prevalent splicing factor gene mutations dominate. In addition, the expression of several mRNA splicing factor genes is changed in lung cancer, thereby changing mRNA splicing patterns [212] (Figure 6b).

#### 3.2.1. U2AF35

The only spliceosome core component that is regularly found mutant in lung cancer is U2AF35, albeit in only approximately three percent of lung adenocarcinomas [191,213,214]]. U2AF35 encodes the small subunit of the U2 auxiliary factor (U2AF). U2AF35 recruits the U2 snRNP to the 3′ ss through binding to consensus AG sequence motifs. Mutations in U2AF35 were studied in detail in hematological malignancies. They occur almost exclusively in specific residues in its two zinc finger domains [215,216,217]. In lung adenocarcinoma, the mutation in the first zinc finger domain appears predominant [214]. Although the role of the U2AF35 zinc finger domains in mRNA binding is not clearly confirmed, mutations in these domains were shown to change the U2AF35 mRNA-binding preference, suggesting that they contribute to guiding the U2 snRNP to the 3′ ss. Mutant U2AF35 recognized 3′ splice sites with a nucleotide immediately adjacent to the AG core consensus sequence that is not normally recognized by wild type U2AF35, causing specifically altered exon usage patterns [216,217]. Interestingly, U2AF35 mutations in cells appear to be heterozygous, suggesting that loss of the wild type protein is not compatible with cell survival [215]. Indeed, when wild type U2AF35 was knocked out in heterozygous NSCLC cell lines, cell proliferation was abrogated completely, confirming that mutant U2AF35 could not compensate for loss of the wild type protein. In contrast, loss of the mutant allele did not compromise cell growth. Hence, U2AF35 mutation appears neither sufficient nor necessary for uncontrolled NSCLC proliferation [213]. Some insight into the functional role of mutant U2AF35 expression in lung adenocarcinoma was derived from a transcriptome-wide analysis of a NSCLC cell line in which mutant U2AF35 was overexpressed [214]. The induced gene set was enriched for genes associated with epithelial to mesenchymal transition (EMT). For a subset of these genes, AS in response to mutant U2AF35 expression was seen. Moreover, mutant U2AF35 overexpression increased invasiveness of the cells, without affecting their proliferation, which is consistent with EMT [214].

#### 3.2.2. RBM Proteins

Mutations are also found in the spliceosomal RNA-binding protein RBM10, in 5–7% of lung adenocarcinomas [191,218]. RBM10 mutations co-occur with mutations in known lung adenocarcinoma oncogenes (KRAS, EGFR, PIK3CA [191]. Wild type RBM10 promotes exon skipping via binding to intronic regions near splice sites [219]. Missense or truncating RBM10 mutations found in lung cancer patients were shown to disrupt RBM10-mediated splicing regulation of the Notch pathway regulator NUMB (Figure 6b). Loss of NUMB exon 9 skipping in RBM10 mutant cells induced production of a Notch-activating and thus pro-proliferative NUMB isoform [220]. Interestingly, RBM10 also suppresses exon inclusion on its own pre-mRNA as well as on the pre-mRNA of its closest paralog RBM5, reducing the expression of these RBPs by promoting NMD [221]. In a subset of lung adenocarcinomas with mutations in the RBM10 gene, the mutations were localized in the splice sites causing the same exon skipping events that normally occur when RBM10 is wild type, resulting in much reduced RBM10 expression by NMD [221]. The RBM5 and RBM6 genes are located on a chromosomal region that is often deleted in lung cancer, resulting in decreased expression of these proteins. Intriguingly, while RBM5, RBM6 and RBM10 exhibited quite distinct effects on mRNA splicing in a global analysis, depletion of RBM5/6 and RBM10 had opposite effects on AS of NUMB. Silencing RBM5 or RBM6 promoted NUMB exon 9 skipping and reduced clonogenic cancer cell growth; whereas silencing RBM10 promoted exon 9 inclusion and clonogenic cell proliferation [220]. Reduced expression of the paralog RBM4 was also found in NSCLC, as well as in other cancer types, and shown associated with a poor prognosis [222]. Modulation of RBM4 expression changed mRNA splicing patterns, with overexpression in cancer cells causing exon skipping at splice sites near RBM4 binding motifs and inhibition of tumor progression. Here, RBM4 was shown to compete for binding to the same cis-acting element on the pre-mRNA with splicing factor SRSF1 [222]. It is likely that loss of other RBM proteins in NSCLC has similar mRNA splicing dysregulating effects.

#### 3.2.3. QKI-5

Another mRNA splicing factor that is associated with dysregulated NUMB splicing and poor survival in NSCLC is QKI [223,224]. While QKI is found mutated in a subset of malignant gliomas, mutations in QKI are not frequently found in other cancers including lung cancer [225]. Nevertheless, QKI expression is often decreased in NSCLC and loss of QKI function can thus be associated with lung cancer [223,224]. The QKI gene encodes at least three different isoforms that are generated by alternative splicing, of which isoform QKI-5 is found predominantly in the cell nucleus and was shown to affect mRNA splicing [226]. A genome-wide analysis of mRNA splicing changes in NSCLC compared to matched non-malignant lung tissues revealed that a large fraction of the identified differential splicing events were targets for QKI, suggesting that QKI is a major regulator of AS in lung cancer [223]. One of the top ranking targets for AS regulation by QKI was NUMB, with reduced expression of QKI in NSCLC causing increased inclusion of NUMB exon 12 [224] (Figure 6b). QKI-5 was shown to bind to two consensus sequence motifs near the 3′splice site of NUMB intron 11, one site upstream in the intron and the other one in exon 12; and to promote exon skipping by competing with SF1 for binding to the BPS [224]. Similar to what was observed for RBM10, QKI expression promoted exon skipping in NUMB to produce an isoform that suppresses Notch signaling, cell proliferation and transformation [224]. The RBM proteins and QKI could thus be considered tumor suppressors. Other interesting targets for AS regulation by QKI in lung cancer are ESYT2 and ADD3, which encode proteins involved in cytoskeleton organization. ESYT2 encodes an endoplasmic reticulum (ER) membrane anchored protein that appears to induce formation of contacts between the ER and plasma membranes, participating in phosphoinositide homeostasis [227]. The ESYT2 gene carries a cassette exon between exons 13 and 14. QKI promotes skipping of this exon, resulting in a short splicing variant, which is the predominant variant in non-malignant lung tissue. NSCLC cells with low QKI expression have higher levels of the ESYT2 long variant with the cassette exon [223]. Knockdown of the long and short ESYT2 variants in NSCLC cells changed cytoskeleton organization and intracellular vesicle localization. From these experiments it could be inferred that the ESYT2 splicing switch due to loss of QKI expression in NSCLC causes a cytoskeletal redistribution that has been related to endothelial infiltration and metastasis; and inhibition of endocytosis, which is typical in highly proliferative tumor cells [223]. The ADD3 protein together with its family member ADD1 forms Adducin oligomers that recruit spectrin to the growing ends of actin filaments and prevent further addition or loss of actin subunits. The activities of Adducin are regulated by phosphorylation and calcium signaling. In epithelial cells, Adducin and spectrin are localized near the plasma membrane at sites of cell-cell contact. ADD3 likely plays a role in the formation and maintenance of cellular structure as well as in regulating cell-cell adhesion and motility [228]. The ADD3 gene contains a cassette exon 14 that is preferentially included in the majority of NSCLC tissues and skipped in normal lung tissue [229]. ADD3 exon 14 skipping was shown to be regulated by QKI-5 [224] and recently binding motifs for QKI-5 near the 3′ end of ADD3 intron 13 were identified [230] (Figure 6b). Exon 14 inclusion in NSCLC cells with low QKI-5 activity results in an ADD3 isoform with a longer tail domain carrying a 32 amino acid insert. The function of ADD3 in cytoskeleton organization suggests that the insert might affect cell proliferation and motility. Indeed, induced ADD3 exon 14 skipping using antisense oligonucleotides in NSCLC cell lines reduced their proliferation and migration, whereas overexpression of the long exon 14 inclusion variant increased these properties [230]. Together, downregulation of QKI-5 expression in NSCLC appears to play a role in tumor invasion and metastasis by dysregulating cytoskeleton remodeling, at least in part through reduced exon skipping in the ESYT2 and ADD3 genes and probably several other genes involved in these processes.

#### 3.2.4. SR and SR-like Proteins

Overexpression of several SR and SR-like proteins, in particular SRSF1, SRSF3, SRSF6 and TRA2β, was observed in lung cancer. The SRSF1 gene was found overexpressed compared to matched normal tissue in approximately 20% of lung cancers [231]. This could be caused by the location of the SRSF1 gene on a genome region that is often amplified in cancer, but it was also shown that SRSF1 expression is elevated in lung cancer through transcriptional activation by MYC and that SRSF1-induced AS events contributed to the oncogenic functions of MYC [232]. Experimental overexpression of SRSF1 in mouse fibroblasts increased anchorage-independent colony growth in vitro and tumor growth in nude mice in vivo; and protected adenovirus E1A-transformed mouse embryo fibroblasts against apoptosis [231]. Together, these observations clearly identified SRSF1 as a proto-oncogene. Analysis of a series of AS events in putative tumor suppressors and proto-oncogenes revealed that modulation of SRSF1 expression levels changed the mRNA splicing patterns of many genes. For one of the AS events, relevance for lung cancer was shown. SRSF1 promotes AS of the RPS6KB1 gene, including three alternative cassette exons between exons 6 and 7 and an alternative splicing event within intron 7 creating a new poly(A) site (Figure 6b). The alternative splice variant encodes a shorter isoform with oncogenic properties. In a panel of human lung tumors, SRSF1 expression levels correlated with the occurrence of the AS event in the RPS6KB1 gene [231]. Experimental knockdown of SRSF1 or the RPS6KB1 splice variant induced by SRSF1 in a NSCLC cell line with high SRSF1 expression inhibited colony formation in vitro; and the former also inhibited xenograft tumor growth in vivo [231]. SRSF3 is overexpressed in many cancers including lung cancer, probably due to gene amplification [233]. Although this has so far not been investigated in NSCLC, studies in other cell types showed that the functional consequence of high SRSF3 expression is stimulation of cell proliferation and neoplastic transformation; and inhibition of cell senescence [233,234]. It is thus likely that SRSF3 also acts as a proto-oncogene in lung cancer. In an analysis of a small set of lung tumors and normal lung tissues, the SRSF6 gene was found amplified and overexpressed in the tumors [235]. Experimental overexpression of SRSF6 in immortal bronchial epithelial cells allowed these cells to form colonies in soft agar and subcutaneous tumors in immune deficient mice, whereas knockdown of SRSF6 in NSCLC cells inhibited these processes [235]. Thus, also SRSF6 is a proto-oncogene in lung cancer. Finally, the gene encoding the SR-like protein family member TRA2β, which is essential for embryonic development, was found amplified in more than 50% of lung squamous carcinomas [218]. In NSCLC, high TRA2β expression correlated with poor differentiation and high proliferation in histological tumor sections; and with clinical stage and poor prognosis [236]. Knockdown of TRA2β in a NSCLC cell line inhibited proliferation and induced apoptosis, suggesting that TRA2β is essential for the survival of these cells [236]. TRA2β binds to ESEs in certain cassette exons to promote exon inclusion [237]. Interestingly, TRA2β promotes inclusion of exon 7 of the SMN genes, which is needed to produce full length functional protein [238]. Since SMN protein as part of the SMN complex is essential to chaperone the assembly of the Sm-snRNA complexes that form the cores of the spliceosome subunits, we hypothesize that TRA2β overexpression in lung cancer cells might perhaps promote efficient mRNA splicing.

For all highly expressed SR and SR-like proteins in cancer, dysregulated mRNA splicing events have been reported that likely contribute to the oncogenic consequences of their overexpression in lung cancer. In addition to AS of RPS6KB1 mentioned above, this includes, e.g., AS of MKNK2 to produce an oncogenic isoform of this kinase; AS of INSR to produce a mitogenic isoform of the insulin receptor; and AS of BIN to produce a variant with lost ability to bind MYC and thus with lost tumor suppressor activity [231,235]. However, overexpression of SR and SR-like proteins causes numerous AS events, suggesting that the consequences of this dysregulation for the biology of lung cancer are much more complex. Notably, since high expression of SR and SR-like proteins has oncogenic potential, these proteins auto-regulate their production by driving AS of their own pre-mRNA, to produce mRNA variants that are targets for degradation by NMD [239]. Apparently, for several of these proteins this surveillance mechanism is dysfunctional or bypassed in cancer.

#### 3.2.5. hnRNPs

The other important class of mRNA splicing factors, the hnRNPs, are also often dysregulated in lung cancer. A comprehensive genomic and transcriptomic analysis of 22 hnRNP genes (not all of which are confirmed regulators of mRNA splicing) in 33 cancer types revealed that many of these genes had frequent mutations and/or were overexpressed in lung adenocarcinoma as well as lung squamous cell carcinoma. High hnRNP expression was associated with a poor prognosis in lung adenocarcinoma, but not squamous cell carcinoma [240]. A paired analysis of tumor and adjacent tissues from patients with NSCLC for expression of several hnRNPs showed that in particular hnRNPA1 expression was frequently increased (i.e., more than 2-fold in 16/21 cases) [241]. Interestingly, a study in mice showed that during lung oncogenesis initially expression of hnRNPA1 and SRSF1, which act antagonistically in splice site selection to regulate AS, rose together in adenomas, but hnRNPA1 increased much further in tumors reaching 6-fold higher protein levels. In addition, in contrast to SRSF1, hnRNPA1 exclusively localized in tumor nuclei. This was associated with a changed mRNA splicing pattern, probably caused by the shifted balance in nuclear hnRNPA1 and SRSF proteins [242]. The hnRNPs participate in multiple processes related to tumorigenesis, such as alternative mRNA splicing, mRNA stabilization, and transcriptional and translational regulation. Therefore, while regulation of AS is the main function of nuclear hnRNPs, the association of high hnRNP expression with cancer is not necessarily explained by dysregulated mRNA splicing. However, a number of studies have clearly shown AS of target genes by hnRNPs in NSCLC cells. For example, hnRNPL induced AS of caspase-9 (CASP9), promoting the production of an anti-apoptotic variant (Figure 6b). Through recognition of an ESS in exon 3, hnRNPL promoted skipping of exons 3–6. The majority of NSCLC tumors exhibited this dysregulated CASP9 mRNA splicing compared to normal lung tissue. Furthermore, experimental modulation of hnRNPL expression regulated the expression of the two CASP9 variants in NSCLC cells, with knockdown abrogating clonogenic growth and xenograft tumor formation [243]. Another study showed that knockdown of the highly homologous hnRNPA1 and hnRNPA2 proteins in NSCLC cells increased skipping of exon 2 or exons 2 and 3 of the interferon regulatory factor 3 (IRF-3) gene (Figure 6b). The two RBPs were shown to bind to an ISE in intron 1. The exon skipping events induced by silencing hnRNPA1/A2 decreased the amount of IRF-3 protein expressed, consequently reducing the expression of downstream interferon-β and interleukin-10 effector proteins [244]. Remarkably, while hnRNPs and SR proteins usually exhibit antagonistic effects, almost identical effects on IRF-3 were observed when SRSF1 was silenced [244]. In the majority of NSCLC tumors, hnRNPA1/A2 and IRF-3 were overexpressed, suggesting that hnRNPA1/A2 may play a role in maintaining an immune suppressive microenvironment in NSCLC tumors [244]. The PPT binding protein hnRNPI that is overexpressed in lung adenocarcinoma and is associated with a dismal prognosis was shown to promote exon 11a skipping in the ENAH pre-mRNA [245]. The ENAH protein is a regulator of actin dynamics that plays an important role in metastasis. High expression of hnRNPI in NSCLC cells was shown to promote migration and invasion, at least in part mediated via AS of ENAH [245]. Finally, the family members hnRNPA1, hnRNPA2 and hnRNPI were shown to regulate the typical metabolic switch in cancer cells from oxidative phosphorylation to aerobic glycolysis known as the Warburg effect. The glycolytic phenotype is regulated by pyruvate kinase M (PKM). Several isoforms of this enzyme are produced through AS of the PKM gene (Figure 6b). During oncogenesis, the PKM1 isoform that is expressed in most adult tissues is uniformly lost in favor of the embryonic PKM2 isoform [246]. The two isoforms are the products of mutually exclusive exon inclusion events, with PKM1 including exon 9 and PKM2 including exon 10. David et al. showed that hnRNPI binds to a sequence motif in the PPT of intron 8 and hnRNPA1/A2 to a sequence immediately downstream of the 5′ splice site of intron 9 [247]. Using RNAi silencing experiments, they showed that hnRNPA1/A2 and hnRNPI cumulatively contribute to exclusion of exon 9 and inclusion of exon 10. The former likely through their known mRNA splicing inhibitory activities and the latter via an unexplained mRNA splicing enhancing activity [247]. The significance of the PKM1/PKM2 switch in NSCLC is illustrated by the association between PKM2 levels and lung cancer progression; and the promotion of lung cancer cell migration, invasion and metastasis by secreted PKM2 [248].

#### 3.2.6. Dysregulated Phosphorylation of Splicing Factors

Nuclear-cytoplasmic shuttling of hnRNPs and SR proteins is regulated by phosphorylation [249,250]. Since the main nuclear function of these proteins is to regulate mRNA splicing, their phosphorylation provides an additional layer of AS regulation. In fact, mRNA splicing patterns can be changed by phosphorylation of splicing factors without any apparent difference in expression of these factors. A good example of this situation is the AS of CASP9 in NSCLC mentioned above [243]. Here, hnRNPL was not overexpressed but hyperphosphorylated in NSCLC. Phosphorylated hnRNPL in contrast to unphosphorylated protein bound to the ESS in CASP9 exon 3, which has a sequence different from the canonical hnRNPL recognition element (Figure 6b). This suggests that the phosphorylation changed the mRNA binding specificity of hnRNPL, or, alternatively, recruited an unidentified accessory protein to aid binding [251]. Furthermore, in dephosphorylated NSCLC cells, there is significant enhancement of binding of hnRNPU to exon 3 of CASP9, which results in inclusion of exons 3-6 to produce the pro-apoptotic CASP9 isoform [252]. Upon AKT-mediated phosphorylation, phosphorylated hnRNPL competes with hnRNPU for the binding site, thereby suppressing exon inclusion [252]. This regulation is further strengthened by binding of only unphosphorylated SRSF1 to an ISE in intron 6 of CASP9 to promote inclusion of the exon 3-6 cassette [253] (Figure 6b). Thus, phosphorylation tightly regulates CASP9 AS at least via inhibition of SRSF1 and stimulation of hnRNPL interaction with the pre-mRNA to promote production of the anti-apoptotic short isoform.

## 4. Therapeutic Targeting of Dysregulated mRNA Splicing in Cancer

Based on the fundamental knowledge of dysregulated mRNA splicing in cancer, there have been many efforts to correct specific AS events in cancer cells, for example using splice variant-specific siRNAs or antisense oligonucleotides. These approaches have been reviewed elsewhere [254]. In this review, we exclusively focus on attempts to modulate the mRNA splicing machinery. In part, this includes discovery of inhibitors targeting specific components of the spliceosome. Additional therapeutic targets in the splicing machinery were identified through unbiased phenotypic screening efforts or by investigations into the mechanism of action of natural compounds with anticancer activity. In this section, several of these studies and discovered therapeutic targets and drugs will be discussed. Highlights are illustrated in Figure 7. Here, a distinction is made between targeting auxiliary proteins that regulate the RNA splicing reaction and targeting core components of the spliceosome.

### 4.1. Target Discovery

#### 4.1.1. Splicing Factor Regulatory Kinases

Kinases represent an important class of targets for the development of new cancer therapeutics. Therefore, target discovery efforts often focus on the kinome. Consequently, it doesn’t come as a surprise that spliceosomal kinases were identified as potential therapeutic targets in cancer cell viability screens. In a kinome-wide siRNA screen on a human pancreatic cell line that is particularly insensitive to apoptosis, the serine/threonine-protein kinase PRP4 homolog (PRPF4B) was identified as an essential gene. Silencing PRPF4B caused an increase in apoptosis in three different pancreatic cancer cell lines [255]. PRPF4B was also found in cell viability dropout screens with pooled short hairpin libraries on lymphoma, colorectal cancer (CRC) and breast cancer cell lines [256,257]. This particular kinase associates with SF1 and U2AF65 [258], which are involved in the initial recognition of splice sites and recruitment of the spliceosome to the pre-mRNA (or complex E formation). Here, PRPF4B modulates alternative splice site selection. Interestingly, PRPF4B was found to be involved in the splicing of the Bcl-x apoptosis regulator and FAS cell surface death receptor genes. PRPF4B overexpression increased the ratio of pro-apoptotic Bcl-xS over anti-apoptotic Bcl-xL and PRPF4B knockdown did the opposite. In contrast, PRPF4B overexpression resulted in increased exon 6 skipping in FAS, giving rise to an anti-apoptotic soluble FAS isoform; and silencing PRPF4B increased expression of the pro-apoptotic transmembrane form of FAS. While these observations suggest contrasting effects on the intrinsic and extrinsic apoptosis pathways, functional experiments showed that the overall result of PRPF4B silencing is induction of programmed cell death [258]. Apart from its role as a survival gene, PRPF4B was found to contribute to drug resistance in ovarian cancer cells [259]. Paclitaxel resistant cell lines were found to overexpress the gene and experimental overexpression in drug-sensitive cell lines induced resistance to paclitaxel, doxorubicin and vincristine. Moreover, knockdown of PRPF4B in multidrug resistant cell lines sensitized the cells to paclitaxel. It was not investigated if these effects were caused by dysregulation of AS or by inhibition of other PRPF4B functions. Nevertheless, it could be concluded that inhibition of PRPF4B activity can be considered for the treatment of chemotherapy-resistant cancer.

#### 4.1.2. Splicing Factors

Many auxiliary spliceosomal components such as hnRNPs and SR proteins were identified as a weak spot in cell viability screens on cancer cells. In a study by Sakuma et al. [260], a murine rectal cancer cell line was transduced with a genome-wide shRNA library and transplanted into the guts of mice, where tumor formation was followed and metastatic tissue was analyzed for depleted transcripts. The authors showed that in metastases the expression of hnRNPLL was decreased, thereby concluding that hnRNPLL is a suppressor of metastasis. Specifically, hnRNPLL was shown to affect splicing of CD44, prohibiting the inclusion of cassette exon v6. This CD44 variant is associated with tumor aggressiveness and metastatic potential. In CRC cells, hnRNPLL was downregulated during EMT, which is in line with its proposed effect on metastasis. Another RNAi screen, in search of regulators of the senescence-associated secretory phenotype (SASP), i.e., a complex combination of pro-inflammatory factors secreted by senescent cells, identified hnRNPI (PTBP1) [261]. SASP is thought to promote a carcinogenic immune response in advanced cancer. In the screen, silencing of hnRNPI in senescent lung fibroblasts inhibited secretion of the cytokines IL-6 and IL-8. This was confirmed in other senescent cells; and silencing hnRNPI reduced secretion of most pro-inflammatory cytokines. Mechanistically, hnRNPI was shown to regulate alternative splicing of EXOC7, thereby promoting SASP. In a xenograft mouse model using mixed squamous cell carcinoma cells and senescent fibroblasts, depletion of hnRNPI inhibited SASP-promoted tumor growth. Interfering with hnRNPI expression in senescent hepatocytes decreased in vivo liver tumor formation in immune competent mice, suggesting that hnRNPI might be a therapeutic target [261]. In glioblastoma multiforme (GBM), hnRNPH was found overexpressed and associated with oncogenesis [262]. hnRNPH was discovered by mutational analysis of putative cis-acting elements in the IG20/MADD gene. This gene encodes pro-apoptotic IG20 and anti-apoptotic MADD variants that are produced by AS. In GBM, mainly the MADD variant is expressed. An ESE was identified in exon 16 and shown to be regulated by hnRNPH. Experimental silencing of hnRNPH caused exon 16 inclusion, switching AS from MADD to IG20, and cell death. Furthermore, hnRNPH was found to control AS of the RON gene, promoting invasion [262]. Thus, hnRNPH is a putative target for treatment of GBM.

The hnRNP counterparts; the SR proteins, were also identified as potential therapeutic targets in cancer. SRSF1 was found to be upregulated in glioma, and this correlated with poor survival [263]. Increased expression of SRSF1 appeared to promote cell survival, probably through guiding alternative splicing of MYO1B towards its oncogenic full-length transcript variant. Silencing SRSF1 decreased tumor growth in vivo and this led to an increased overall survival in mice [263]. SRSF1 was also found to confer resistance to gemcitabine in pancreatic cancer cells through promoting alternative splicing of MKN2 into the MNK2b transcript variant [264]. Gemcitabine treatment induced SRSF1 expression and thus MKN2b expression. This variant produces an oncogenic protein isoform that promotes phosphorylation of eIF4E. The latter protects pancreatic cells from gemcitabine treatment and is correlated with poor prognosis in pancreatic cancer. Silencing SRSF1 restored splicing of MNK2 to the canonical MNK2a variant, led to decreased phosphorylation of eIF4E and increased the sensitivity of pancreatic cancer cells to gemcitabine treatment [264].

#### 4.1.3. RNA Helicases

Also, RNA helicases in the spliceosome might be of therapeutic interest in cancer. In a focused screen using an shRNA library silencing spliceosome-related genes, ATP-dependent RNA helicases 39A and 39B (DDX39A and DDX39B) were found to be involved in the splicing of the androgen receptor (AR) in prostate cancer [265]. Knockdown of DDX39B resulted in a decrease of specifically the ARv7 variant associated with poor prognosis in this type of cancer, and this was further enhanced when simultaneously silencing its paralog DDX39A. This can be explained by the fact that these helicases have redundant functions and that DDX39B is likely to suppress DDX39A [265].

#### 4.1.4. U2 snRNP Components

Aside from auxiliary spliceosomal proteins, components of the core spliceosome were also identified as potential cancer therapeutic targets. Several targets were found in the U2 snRNP. The identification of these targets was supported by the discovery of natural products with anticancer properties that were found to target the SF3b complex (see below) and by the documentation of frequent mutations in the U2 snRNP genes SF3B1 and U2AF35 in myeloid neoplasms [266], which has fueled research into this part of the spliceosome. For example, U2 snRNP-specific SF3A1 and SNRPA1 were identified as survival genes in multiple myeloma [267] and we found that silencing SF3A3 was selective lethal in NSCLC [268]. In a genome-wide RNAi screen on GBM stem cells and untransformed neural stem cells and fibroblasts, it was found that the SF3b-specific PHF5A protein was required for GBM stem cell survival and tumor growth through regulating exon recognition of essential genes [269]. Inhibition of the SF3b complex through small molecules targeting SF3B1 resulted in splicing defects in cell cycle genes, specifically in GBM cells [269]. In another genome-wide RNAi screen, Grohar et al. [270] found that SF3B1 and SF3A1 were required for proper processing of EWS-FLI1 transcripts in Ewing sarcoma. In this disease, a translocation event occurs in which the 5′ end of the EWSR1 gene and the 3′ end of the FLI1 gene are fused, expressing oncogenic EWS-FLI1 transcripts [271]. Silencing SF3B1 or SF3A1 reduced EWS-FLI1 protein activity and Ewing sarcoma cell survival; and inhibition of SF3B1 with a small molecule resulted in mis-splicing of EWS-FLI1 and loss of Ewing sarcoma cell viability [270]. Recently, the U2 snRNP-related factor U2AF65 was found overexpressed in NSCLC and expression levels correlated with tumor progression. Silencing U2AF65 decreased cell viability in NSCLC cell lines [272]. The utility of targeting SF3B1 to treat cancer is actively investigated using a variety of SF3B1-targeted compounds (see below). While selective lethality on cancer cells and anticancer activity of these compounds were shown, whether SF3B1 is a truly cancer-selective target is somewhat controversial. For example, we found that SF3B1 silencing induces profound AS in the transcriptomes of malignant as well as non-malignant lung cells [273] and is lethal to both cell types [268,274]. Moreover, in an elegant study Zhou et al. [275] raised serious questions concerning the use of SF3B1 inhibitors. Using gene editing knock-in techniques, they introduced by homologous recombination sequences into endogenous SF3B1 alleles encoding an in frame N-terminal tag that can be targeted for proteasomal degradation when a specific ligand is added. This way, they could induce individual depletion of wild type and mutant SF3B1 proteins in cancer cell lines. Surprisingly, depletion of wild type SF3B1 in homozygous or heterozygous cell lines was lethal, whereas depletion of mutant SF3B1 in heterozygous cell lines was not. Interestingly, depletion of mutant SF3B1 did alter AS patterns, including a reversal of AS events known to correlate with mutant SF3B1 expression, confirming functional inactivation of the mutant protein. Hence, SF3B1 appeared an essential gene and SF3B1 mutations in cancer cells, at least the ones investigated, were not essential for cancer cell growth. This indicated that inhibition of wild type SF3B1 is likely cytotoxic, also in cells lacking oncogenic SF3B1 mutations; and, since all SF3B1 mutant cancers are heterozygous, selective inhibition of mutant SF3B1 might not have strong therapeutic effects.

#### 4.1.5. Other Spliceosome Structural Components

Also certain components of the U4/U6.U5 tri-snRNP represent vulnerabilities of the cancer cell. For example, in an siRNA screen on triple-negative breast cancer cell lines, silencing of PRPF8 and pre-mRNA processing factor 38A (PRPF38A) were shown to confer lethality and disrupt mitosis. Knockdown of PRPF8 and PRPF38A led to increased intron retention in transcripts from genes involved in protein homeostasis, mitosis and apoptosis, potentially explaining the observed phenotypic effects [276]. In a synthetic lethal screen with siRNAs targeting proteins with deubiquitinase or ubiquitinin-like motifs, PRPF8 and the complex B-associated protein ubiquitin-like protein 5 (UBL5) were found to promote resistance to Bcl-2 inhibitor ABT-737 in neuroblastoma cells, through AS of induced myeloid leukemia cell differentiation protein MCL1 into an anti-apoptotic short transcript variant. Therefore, targeting these spliceosomal components could be of interest in re-sensitizing MCL1-dependent neuroblastoma cells to Bcl-2 inhibitors [277]. PRPF4, which is part of the U4/U6 di-snRNP, was found to be overexpressed in breast cancer cell lines. Silencing PRPF4 resulted in apoptosis, suppression of EMT and reduced migration and invasion of breast cancer cells. In addition, knockdown of PRPF4 resulted in decreased p38MAPK signaling [278]. Although it remains to be confirmed if these effects were caused by inhibition of the spliceosome or by a possible other function of PRPF4, the observations suggest that PRPF4 could be a useful target to treat cancer.

The NTC and NTR complexes are also core components of the spliceosome. These are not associated with a specific snRNP, but rather with the complex B as a whole. The NTR protein SYF2 [279] was, together with the zinc finger RNA-binding splicing factor ZRANB2, identified as a putative therapeutic target in a small siRNA screen silencing 19 spliceosome proteins that were found overexpressed in doxorubicin-resistant breast cancer cells. Depletion of SYF2 and ZRANB2 reduced cell survival in the presence of doxorubicin, suggesting that they contributed to the resistance to this drug. SYF2 was shown to control AS mainly by regulating alternative 3′ss usage, while ZRANB2 primarily regulated cassette exon inclusion. Three AS events were identified and validated that were regulated by both mRNA splicing factors, i.e., cassette exon inclusions in the MRPL55, MAST2 and ECT2 genes. It was furthermore shown that the exon 5 inclusion variant of ECT2 increased tumor growth of doxorubicin-resistant breast cancer cells and resistance to chemotherapy in vivo, suggesting that this AS event contributed to the resistance induced by SYF2 or ZRANB2 overexpression [279].

#### 4.1.6. Sm Proteins

Finally, the Sm proteins, which are present in all snRNPs and play a crucial role in cytoplasmic snRNP core assembly and nuclear import allowing further nuclear snRNP maturation and spliceosome assembly, were identified to be cancer-selective therapeutic targets. This is perhaps surprising, as it could be postulated that as Sm proteins are such an essential and integral part of the spliceosomal machinery, targeting these proteins would lead to a complete collapse of this apparatus. This would in theory be intolerable not only to cancer cells, but also to healthy, non-malignant cells. However, this appears not to be the case. Sm proteins were found to be overexpressed in lung, breast and ovarian cancers [274,280]. Silencing Sm genes decreased cell viability in NSCLC cells, breast cancer cells and melanoma cells [268,274,280], whereas this had little effect on the viability of non-malignant lung fibroblasts and epithelial cells [274] or breast epithelial cells [280]. These observations make Sm proteins particularly interesting targets for treating cancer. Interestingly, knockdown of Sm genes induced apoptosis in NSCLC cells [274], whereas breast cancer cells exhibited a deregulation of the mTOR pathway and induction of autophagy [280]. Thus, while the induction of cell death upon silencing of Sm genes appears a universal response in cancer cells, the type of cell death that is induced seems cancer type or context dependent. In NSCLC cells, silencing SmD3 resulted in the reversal of the lung cancer-specific alternative splicing variant of the ADD3 gene (with retained exon 14) to the non-malignant variant with skipped exon 14 [274]. As discussed above, this AS event is regulated by QKI-5. Hence, apparently silencing of Sm genes could correct a cancer-specific mRNA splicing dysregulation that is not caused by Sm overexpression. It is tempting to speculate that targeting Sm genes could be a viable strategy to correct dysregulated mRNA splicing in general. Obviously, this awaits experimental verification. Silencing SmD3 also changed the AS pattern of p53 regulator MDM4; and increased the transcriptional activity of p53 tumor suppressor protein in p53 wild type NSCLC cells [268]. In fact, silencing many components of the spliceosome increased p53 activity and this was most prominent for the Sm proteins [268]. Furthermore, we recently showed that silencing SmD3 had relatively subtle effects on AS in NSCLC cells when compared to silencing SF3B1 [273]. Among 133 AS events that were unique for silencing SmD3, an exon skipping event in the proteasome subunit beta 3 (PSMB3) gene was most prominent. Silencing of all seven Sm genes induced this AS event in NSCLC cells, but not significantly in lung fibroblasts. The AS switch produced a transcript that is predicted to be a target for NMD and thus to cause a decrease in PSMB3 expression. Loss of the long transcript was cytotoxic and could therefore contribute to the cancer-selective lethal phenotype of Sm gene silencing [273]. Interestingly, in a computational analysis of whole genome siRNA screens on lung cancer cell lines that identified mRNA splicing as a major category of lung cancer vulnerabilities, a subset of cell lines was found particularly sensitive to loss of the U6 snRNP-specific Sm protein counterparts—the LSm proteins [281]. This predicted sensitivity awaits biological validation. Finally, PRMT5, which is involved in the assembly of the Sm ring, was identified as a cancer-specific vulnerability in malignant glioma [282]. The gene emerged as the highest confidence hit in a pooled in vivo shRNA screen targeting epigenetic regulators in intracranial mouse GBM tumors. Downregulation of PRMT5 resulted in decreased glioma cell proliferation and inhibition of tumor formation. Treatment of glioma cells with a PRMT5 inhibitor led to cell cycle arrest and senescence in vitro. In vivo, the inhibitor appears to be incapable of crossing the blood-brain barrier (BBB) in healthy brain, but in mouse studies it accumulated in GBM, where the BBB is disturbed. Sensitivity of GBM cells to this particular inhibitor was accompanied by increased inclusion of so-called detained introns, a specific form of retained introns that are insensitive to NMD and remain in the nucleus. An increase of detained introns leads to a decrease of functional protein isoforms. This was predominantly observed for proliferation genes, such as Aurora kinase B, which could explain the observed cell cycle defects upon PRMT5 inhibition [282].

Overall, these studies show that there are multiple approaches to targeting the spliceosome for cancer therapy, either as a direct treatment or to sensitize cancer cells to chemotherapeutic agents. Potential targets include auxiliary components such as kinases, hnRNPs, SR proteins and RNA helicases; as well as spliceosomal core components, such as members of the U2 snRNP or U4/U6.U5 tri-snRNP. In particular the Sm proteins and proteins involved in the assembly of the Sm protein ring seem to have great therapeutic potential. Several studies suggest that these targets represent cancer-selective vulnerabilities, as opposed to much investigated targets such as the U2 snRNP specific component SF3B1.

### 4.2. Drug Development

Drug discovery is typically done in two different ways. One method that is often used is by phenotypic screening of chemical compounds. Only when candidates with the desired properties are identified, mechanism of action studies are performed to reveal the molecular targets and pathways of these compounds. In view of the relevance of dysregulated mRNA splicing in cancer, many phenotypic screens for molecules that change AS in cancer were done and candidate drugs were identified. For most drugs identified in these screens, the target molecules in the spliceosome are unknown. In addition, phenotypic screens for compounds that are lethal to cancer cells sometimes identified molecules that were later shown to target the spliceosome. The second method is target-based drug discovery. The identification of specific targets in the mRNA splicing machinery with putative therapeutic utility discussed above has motivated scientists to initiate this second type of drug discovery efforts. These two distinct types of drug discovery studies and the further development of candidate drugs will be discussed separately below.

#### 4.2.1. Phenotypic Screening to Discover Spliceosome Targeting Drugs

In search of spliceosome inhibitors, two different types of high-throughput screens were done. In the first type of screens, the effect of compounds on splicing of a reporter pre-mRNA in cells or cell extracts was investigated. In the second type of screens, the effect of compounds on the assembly of the spliceosome onto a pre-mRNA template was studied. Using such screens, many different compounds that inhibit different stages of spliceosome assembly and mRNA splicing were identified.

#### 4.2.2. Screens for Inhibition of pre-mRNA Splicing

In an example of the first type of screen, a reporter cell line was used that carries a luciferase gene with an intron designed such that only the unspliced transcript upon escape from the nucleus can be translated into functional reporter protein [283]. Reporter cells were treated with a library consisting of approximately 8000 natural and synthetic compounds. A limited number of compounds specifically increased luciferase activity in the cells, one of which clearly stood out. This compound, the biflavinoid isoginkgetin (IGG), a natural product found in plants, also inhibited splicing of several endogenous pre-mRNAs in human cells. Treatment of cells with IGG was reversible cytostatic. Analysis of spliceosome assembly in the presence of IGG demonstrated an accumulation of complex A spliceosomes, suggesting that the compound inhibited the transition from A to B, i.e., recruitment of the U4/U6.U5 tri-snRNP [283]. Cell cycle analysis of colorectal and ovarian cancer cell lines treated with IGG revealed a profound inhibition of DNA synthesis, arresting cells in S phase. This reversibly inhibited progression through the cell cycle [284]. Following up on the identification of IGG, four other biflavinoids were compared to IGG for their effects on the splicing of pre-mRNA substrates [285]. While three of the newly tested compounds exhibited inhibitory activity on the mRNA splicing reaction, only one equaled or even exceeded the effects of IGG. This compound, hinokiflavone (HKF), induced AS of several tested transcripts, with differential effects in different cell lines. Its effect included promoting exon 2 skipping of MCL1, increasing the ratio of pro-apoptotic over anti-apoptotic MCL1 isoforms; and dose-dependent induction of cell cycle arrest and cell death [285]. Administration of HKF in colorectal [286] and breast [287] carcinoma mouse models reduced tumor cell growth, demonstrating in vivo anti-cancer effects. Mechanistically, HKF appeared to inhibit SUMO protease activity, causing cells to accumulate hyperSUMOylated proteins. An unbiased analysis revealed that 10 of 22 lysine residues with strongly increased SUMO modification upon HKF treatment were located in six proteins of the U2 snRNP, with most dramatic effects on PRPF40A [285]. PRPF40A is associated with U2 snRNP, interacts with SF1 and U2AF65 and is present in spliceosomal complexes A and B. Its yeast orthologue was proposed to be involved in cross-intron bridging (i.e., bringing the splice sites at both ends of the intron in closer proximity); however, for human PRPF40 the exact role remains unclear [258]. Aside from U1 and U2 snRNP-associated proteins, PRPF40A was also found to interact with SART3, which is associated with the U4/U6.U5 tri-snRNP [288]. Since HKF inhibits deSUMOylation of U2 snRNP proteins and the transition of spliceosome complex A to B, it could be speculated that deSUMOylation of U2 snRNP proteins is perhaps required for recruitment of the U4/U6.U5 tri-snRNP.

In another study, a reporter construct was used that carries a cassette exon. Inclusion or exclusion of this exon changes the reading frame, producing alternative transcripts encoding either green or red fluorescent protein. With a primary interest in studying dementia with parkinsonism, the cassette exon used was derived from the MAPT gene, the inclusion of which is associated with this disease [289]. The cells were treated with two libraries, together comprising 1440 known drugs. The ratio of red and green fluorescence intensities was used to screen for compounds that change the mRNA splicing pattern. The screen was troubled by differences in the stability of the alternative splicing variants. Nevertheless, several compounds were confirmed to influence AS, either promoting or suppressing exon inclusion. A few of these presumably inhibited mRNA splicing per se, whereas others influenced AS. Three of these drugs were tested for their effects on AS of multiple exons in human cancer cells. Each compound had distinct but overlapping effects, suggesting that they were acting through different molecular mechanisms [289]. Although this screening method has its limitations, it could be a useful strategy to discover drugs that counteract dysregulated mRNA splicing patterns in lung cancer, using a similar construct with a NSCLC-specific cassette exon, such as e.g., from the NUMB, ADD3, or caspase-9 genes.

A different strategy used to screen for inhibition of mRNA splicing exploited the fact that upon completion of the mRNA splicing reaction a unique molecular signature of EJC proteins remains bound near to splice junctions. Biotinylated synthetic pre-mRNA substrate derived from human adenovirus was mixed with nuclear extract from HeLa cells to allow in vitro mRNA splicing. Subsequently, the mRNA was captured in assay plates and subjected to an ELISA with an antibody against one of the EJC proteins. This assay was employed in a high-throughput setting to screen a collection of 2100 annotated bioactive molecules for compounds that reduced EJC binding and thus mRNA splicing [290]. A selection of hit compounds was validated by showing that they significantly reduced the ratio of spliced over non-spliced mRNA. The most potent mRNA splicing inhibitor that was discovered was a 1,4-naphthoquinone molecule. Structure and activity probing resulted in the identification of several structurally similar compounds with a 1,4-naptho- or 1,4-heteroaryl-quinone scaffold that were more potent in inhibiting mRNA splicing. The most potent compound termed BN82685 was shown to inhibit the mRNA splicing reaction at the second transesterification step, thereby preventing the release of the intron lariat and exon ligation [290]. This compound was previously shown to inhibit tumor growth in pancreatic cancer xenografts, presumably through the inhibition of Cell Division Cycle 25 (CDC25) phosphatases [291]. In line with this, another study showed that BN82685 impaired mitotic spindle assembly and inhibited proliferation of CRC cells in vitro [292]. Possibly, part of its anticancer effect could be explained by its previously unrecognized effect on mRNA splicing.

Effenberger et al. [293,294] also used a screening method where the effect of compounds on mRNA splicing was analyzed in nuclear extract from HeLa cells using a synthetic pre-mRNA derived from the adenovirus major late transcript as substrate. Splicing efficiency was quantified by RT-qPCR for an exon-exon junction. A high-throughput screen done with a library of approximately 3000 small molecules identified 6 compounds that consistently inhibited in vitro mRNA splicing, 3 of which exhibited clear dose-dependent effects [294]. These three compounds interfered with the spliceosome at different steps of the splicing reaction. The first one is tetrocarcin A (TCA), which has previously shown anti-tumor effects as a Bcl2 inhibitor [295]. Treatment with TCA resulted in accumulation of a spliceosome A-like complex, suggesting that it stabilizes this complex or inhibits transition to the next assembly stage. Triple-negative breast cancer cells (TNBCs) treated with TCA showed decreased viability in vitro and in an in ovo xenograft model. Efficacy was accompanied by downregulation of the junctional adhesion molecule-A and caspase-dependent apoptosis [296]. The second compound is indole derivative NSC635326 with no known biological activity, which caused accumulation of the very early complex E. Therefore, both TCA and the indole derivative interfere with early spliceosome assembly [294]. The third compound is naphthazarin derivative NSC659999, that also showed anti-cancer properties [295]. This compound inhibited the formation of spliceosome C complexes. It did, however, not change the kinetics of complex A or complex B formation, suggesting that it interferes with a factor involved in late spliceosome assembly [294]. Using the same method, a second library was screened consisting of over 5000 fractions of crude lysates derived from a diversity of marine bacteria harvests, comprising an estimated ~50,000 natural products [293]. Different fractions were found to inhibit mRNA splicing, stalling spliceosome assembly at stages after complex A formation. From active fractions, 6 distinct compounds were purified that inhibited the spliceosome. One active compound, produced by bacteria from three distinct locations, was characterized as N-palmitoyl-L-leucine (NPLL). Synthetic NPLL was shown to inhibit late spliceosome assembly, accumulating in a B-like complex; and to inhibit splicing of a panel of tested pre-mRNA substrates [293].

Another reporter construct that was designed to screen for mRNA splicing modulators consists of the luciferase gene with an intron that confers efficient splicing. In case of intron retention, intronic in-frame stop codons produce truncated inactive luciferase. Hence, mRNA splicing inhibitors reduce luciferase activity in reporter cells carrying this construct [297]. The assay was used to screen over 23,000 commercially available compounds. Molecules that reduced luciferase activity were rescreened on cells expressing intronless luciferase, to filter away compounds that inhibit the enzyme or reduce gene expression through other means than inhibition of mRNA splicing. Three molecules that are in wide clinical use were further investigated, i.e., clotrimazole, flunarizine and chlorhexidine. When tested on endogenous genes, the compounds exhibited unique patterns of mRNA splicing inhibition. Clotrimazole and flunarizine were stronger inhibitors of constitutive splicing, whereas chlorhexidine was more effective in modulating alternative splicing. The observation that chlorhexidine modulated the splicing of three pre-mRNAs that are known to be regulated by SR proteins, pointed at a possible role of these splicing factors. Chlorhexidine was found to decrease phosphorylation of several SR proteins, through inhibition of CLKs [297]. Yet another luciferase-based reporter construct consisted of the luciferase gene with an interrupting insert of intronic and exonic (cassette exons 4, 10 and 11) sequences of the MDM2 gene. Only when mRNA splicing is inhibited, resulting in skipping of the MDM2 exons, an uninterrupted functional luciferase open reading frame is produced that can be detected by measuring cellular luminescence [298,299]. Using this readout, screens were done with two libraries of over 5000 bioactive molecules [299] and over 2000 post-phase I anticancer drugs [298], respectively. This identified several known cyclin-dependent and cdc-like kinase inhibitors that were not before shown to modulate mRNA splicing. Three hit compounds including milciclib were screened against a panel of kinases known to regulate AS. This revealed potent inhibition of CLKs and dual specificity tyrosine-phosphorylation-regulated kinases (DYRKs). In addition, the compounds were shown to reduce the phosphorylation of SR and SF3B1 proteins [298]. While the mechanism of mRNA splicing inhibition by the compounds has not yet been resolved, this probably involves the observed inhibition of mRNA splicing factor phosphorylation and suggests that modulation of mRNA splicing contributes to the anticancer activity of the three anticancer drugs.

#### 4.2.3. Screens for Stalling of Spliceosome Assembly

The second type screens measure binding of spliceosomal proteins to a pre-mRNA substrate. Depending on the choice of protein used for readout, specific steps of the splicing process with its dynamic recruitment and release of spliceosome components can be monitored. The readout assay uses nuclear extracts of cells expressing a tagged spliceosome protein that can be detected with a specific antibody, to quantify binding onto an immobilized pre-mRNA substrate in an ELISA. The spliceosome protein that was chosen for use in the reported screens is the ATP-dependent RNA helicase DDX41. DDX41 is recruited when the catalytically active complex C is formed, however, its exact function within the spliceosome is unknown. The pre-mRNA substrate in the assay consists of a 5′ exon and a BPS and PPT-containing intron, but lacks a 3′ splice site and exon. Consequently, the spliceosome can assemble onto the intron, but not complete the splicing reaction and disassemble. Hence, if complex C is formed, tagged DDX41 remains bound to the immobilized pre-mRNA substrate and is detected in the ELISA [300]. The assay was used to screen chemical libraries of ~30,000 [300], more than 70,000 [301] and ~170,000 [302] compounds, respectively, for inhibitors of spliceosome assembly. Confirmed hits were validated based on their capacity to inhibit splicing of adenovirus-derived pre-mRNA. In the first screen, four compounds were identified, including psoromic and norstitic acid [300]. Studies with psoromic acid showed that complex formation was stalled at different stages depending on the compound concentration; and that at a high concentration, at which mRNA splicing was inhibited by ~90%, the spliceosome accumulated mainly as Bact complexes [300]. The second screening campaign consisting of a primary discovery screen, secondary confirmation screens, hit stratification and iterative analysis of analogues; identified 4 compounds that consistently inhibited mRNA splicing in different assays, one of which was further characterized [301]. This compound, madrasin, stalled spliceosome assembly at the complex A, precluding progression to complex B. Madrasin changed the splicing pattern of a panel of tested pre-mRNAs, with MCL1 exon 2 skipping as the strongest effect observed. Cells treated with madrasin showed inhibition of DNA synthesis, but not of RNA transcription. Cell cycle arrest was induced sooner and at lower madrasin dose than needed to inhibit DNA synthesis or to cause exon skipping in the analyzed genes. Therefore, it is not clear if the cytostatic effect of madrasin is caused by AS [301]. In the third screen, eight compounds were confirmed to inhibit mRNA splicing. One of these, termed cp028, was selected for further validation. It was shown to inhibit transition of complex A to complex B, as well as transition of B to Bact. The latter assembly stage was further characterized as a step after displacement of U4 snRNA and U4/U6-specific proteins, but prior to release of the B-specific and LSm proteins and incorporation of the NTC [302]. For none of the inhibitors identified in these screens the spliceosomal targets are known. It is also not known yet if these compounds or derivatives identified by structure activity relationship (SAR) correlations will be useful as candidate anticancer drugs, or that they will merely be tools to study the mRNA splicing process.

#### 4.2.4. Hypothesis-Driven Identification of Spliceosome Inhibitors

Sometimes, candidate drugs that influence AS are not identified in unbiased screens, nor designed to target specific spliceosome proteins, but are developed on the basis of a rational hypothesis. Because a previous study had shown that small changes in the pH of the culture medium could modify the AS pattern of a pre-mRNA in human cells [303], Yuo et al. [304] hypothesized that inhibitors of the plasma membrane Na^+^/H^+^ channel could modulate AS. Aiming to develop a treatment for spinal muscular atrophy (SMA), they investigated if the Na^+^/H^+^ channel inhibitor 5-(N-ethyl-N-isopropyl)-amiloride (EIPA) could promote exon 7 inclusion in the SMN2 mRNA. Treating cell lines from SMA patients with EIPA increased exon 7 inclusion and SMN protein production. EIPA-treated cells exhibited increased nuclear levels of SRSF3, but not SRSF1 and hnRNPA1 [304]. However, when EIPA and the approved diuretic analogue amiloride were tested for their effects on AS in leukemic cells, only the latter induced pro-apoptotic splicing switches in the Bcl-x and HIPK3 genes, suggesting that this effect was unrelated to modulation of the pH [305]. Amiloride changed the expression levels of SR and hnRNP family members, as well as the phosphorylation of SR proteins, but quite differently than was reported for EIPA. Thus, the discovery of amiloride as a potential anticancer drug targeting the mRNA splicing machinery was serendipitous. While amiloride induced cell cycle arrest and apoptosis in leukemic cells [305], effective concentrations were too high to translate its use into the clinic. Therefore, an elegant study employing computational drug design identified potential molecular targets (SNRP70 and hnRNPI) of amiloride and this aided the development of amiloride derivatives with optimized docking properties onto the putative targets [306]. One of these derivatives; BS008, was found to induce cell cycle arrest and apoptosis in hepatocellular carcinoma cells, induce alternative splicing of apoptotic genes and decrease tumor growth in a hepatocellular carcinoma mouse model. Previously seen effects of amiloride on SR and hnRNP proteins were confirmed and extended in this study, together suggesting that amiloride-like compounds modulate AS in cancer cells via a diversity of effects on mRNA splice factors, regulating kinases and histone modifications [306].

#### 4.2.5. Discovery of Inhibitors of the U2 SF3b Complex

Several studies focused on the identification of specific inhibitors for spliceosome proteins that are considered relevant in cancer. Of all specifically targeted spliceosomal components, the U2 SF3b complex is by far the most extensively studied. Many compounds have been isolated and synthesized that are described to inhibit the SF3b complex and to display anti-cancer activity. Historically, screens for natural products with antitumor activity led to the identification of compounds in Pseudomonas and Streptomyces bacteria that were subsequently shown to target the SF3b complex. Paradigm examples of active compounds identified in these species are FR901464 isolated from a Pseudomonas species and pladienolide B (PB) and herboxidiene (HBD) isolated from two different Streptomyces species, respectively (reviewed in [307,308]). Following the discovery of these molecules, many synthetic analogues with improved properties for use as drugs were developed. These include, e.g., the FR901464-derivatives spliceostatin A (SSA) and meayamycin B (MB); and a new class of compounds designed on the basis of a hypothetical consensus pharmacophore model derived from molecular overlays of FR901464 and PB, the sudemycins. In addition, a natural analogue of HB, RQN-18690A, was isolated from another Streptomyces species in a screen for inhibition of endothelial cell motility [309]; and Jerantinine A (JA) that was isolated from the leaves of a plant in search of cytotoxic alkoids, was only later demonstrated to target the spliceosome, by combined functional screening with an RNAi library and proteomics analysis to identify genes that are required for JA activity [310].

An interesting approach taken to expand the panel of available compounds, exploiting pharmacological knowledge of active compounds and the versatility of what nature has to offer was taken by Liu et al. [311]. They used a genomics-guided strategy to identify analogues of FR901464 in a gram-negative bacterial species. After decoding the genetic basis for the biosynthesis of FR901464 in Pseudomonas bacteria, genomic mining in the gram-negative bacteria revealed a highly homologous gene cluster. An extract from bacteria grown under fermentation conditions at which the regulatory gene in this cluster was expressed was analyzed by HPLC to identify three compounds with similar characteristics. These molecules were more stable than FR901464 and one of these, called thailanstatin A, inhibited mRNA splicing with similar efficacy as FR901464 and exhibited only slightly lower antiproliferative effects on cancer cell lines in vitro [311]. To improve delivery of thailanstatin A to cancer cells, it was coupled to an anti-Her2 monoclonal antibody [312]. These antibody-drug conjugates exhibited clear antigen-dependent cytotoxicity on a panel of cancer cell lines, induced AS of CDKN1B; and inhibited tumor growth in a Her2-positive gastric cancer xenograft mouse model [312]. Furthermore, a large number of thailanstatin analogues were synthesized, several of which demonstrated exceptionally strong cytotoxicity in vitro [313]. These molecules could perhaps be developed into potent antitumor agents.

SF3b inhibitors bind to the SF3b complex in its open conformation, thereby interfering with BPS recognition and prohibiting the transition of complex A to an activated spliceosome [309,314,315,316,317,318,319,320]. While the exact targets of the inhibitors are not always known, the three structurally distinct compounds from which most SF3b inhibitors are derived (i.e., SSA, PB and HBD) were shown to compete for binding to the same site on SF3B1 and to induce a conformational change in this protein [321]. Mutations in residues that are part of the BPS binding pocket of SF3b complex proteins, such as SF3B1^R1074H^ and PHF5A^Y36C^ confer resistance to the PB analogue E7107 [316,322], HBD and SSA [322]. Weak BPSs were shown to be more sensitive to interference by SF3b inhibitors [323]. The consequence of the interference with BPS recognition is mainly differential 3′ ss usage resulting in intron retention [324,325,326]. However, the PB analogue H3B-8800 appears to have a preference for introns with higher GC-content and weaker BPS as compared to introns that are retained by E7107 [324], implicating distinct pre-mRNA sequence-specific property differences between these two closely related SF3b inhibitors. The sequences on the 5′ side of the BPS influence drug response; specifically, a longer PPT results in decreased drug sensitivity [325]. Unspliced pre-mRNA upon SF3b inhibitor treatment accumulates in the nucleus or is transported to the cytoplasm for NMD [327]. Interestingly, SSA, HBD and PB were found to not only inhibit SF3B1 functions in complex A formation and stabilization, but when these steps were experimentally bypassed to also interfere with a later function of SF3B1 in the catalytic spliceosome [321].

Some SF3b inhibitors were reported to act via a different mode. Sudemycin E was found to disturb the interaction between U2 snRNP and nucleosomes by binding to SF3B1, causing dissociation of the U2 snRNP from chromatin. This appeared to interfere with the ability of U2 snRNP to maintain histon H3 lysine 36 trimethylation (H3K36me3) at actively transcribed genes [328]. The plant indole alkaloid JA stabilized SF3B1 protein in cancer cells and disrupted the interaction of SF3B1 and SF3B3 with nucleosomes [310]. The SF3b inhibitory activities of sudemycin E and JA are not necessarily distinct from those of other SF3b inhibitors, as their activities may overlap and reported inhibitor characteristics are probably not complete.

SF3b inhibitors were shown to activate apoptosis in a wide range of cancer cell lines, through AS of apoptotic genes such as MCL1 [325,329,330,331,332,333], caspases 2 and 9 [334] and BCL2L1 [330,333]. E7107, PB and MB treatment decreased cell viability in diffuse malignant peritoneal mesothelioma (DMPM) cells. Treatment of DMPM-bearing mice with E7107 resulted in decreased tumor growth and increased survival, accompanied by alternative splicing of the aforementioned genes [333]. In line with this, E7107 was shown to sensitize chronic lymphocytic leukemia (CLL) cells to Bcl-2 inhibitor venetoclax. This was confirmed in an in vivo CLL mouse model [335]. Combined treatment of CLL cells with SSA and Bcl-2 inhibitors increased sensitization of these cells to apoptosis [331]. Additionally, combined treatment of HPV16-positive HNSCC [336] and NSCLC [329] cells with MB and Bcl-2 inhibitor ABT-737 induced apoptosis and overcame resistance to single agent treatment.

Cell cycle arrest has also been observed in response to SF3b inhibition, for instance through the expression of a truncated CDKN1B isoform, which inhibits CDK2 activity [309,318,319,337]. Furthermore, AS of MDM2 [334,338] and proteasomal genes [339] has been reported. In the latter case, H3B-8800 was shown to synergize with proteasome inhibitors to decrease T-cell acute lymphoblastic leukaemia cell growth in vitro [339].

Interestingly, the safety, pharmacokinetics and pharmacodynamics of the PB analogue E7107 were assessed in two phase I clinical studies in patients with solid tumors [340,341]. The two studies reported similar results. E7107 transiently inhibited mRNA splicing in peripheral blood cells, either shown by analyzing a selection of target pre-mRNAs [340] or by a global analysis of intron retention [341]. Antitumor responses were limited, not at all fulfilling expectations from preclinical models. Importantly, in both studies, dose-limiting toxicities were observed. The most remarkable toxicity was vision loss, which led to study termination and discontinuation of further development of this compound. While the underlying mechanism for the vision loss remains to be resolved, these studies advocate prudence in developing treatments based on general inhibition of mRNA splicing.

#### 4.2.6. Discovery of Inhibitors of Later Spliceosome Assembly and Activation Steps

Although more scarce, there have also been efforts to design drugs that target later stages of spliceosome assembly. Diouf et al. [342] developed an assay for binding of SNU13 to the U4 snRNA. This binding is required for subsequent recruitment of other U4 proteins and thus crucial for complex B spliceosome assembly. The assay is based on fluorescence resonance energy transfer between a fluorophore labeled U4 snRNA fragment and indirectly fluorophore labeled recombinant SNU13 protein. The assay was used to screen more than 10,000 bioactive compounds. Topotecan and other camptotecin derivatives were among the most potent inhibitors. These compounds are topoisomerase I (TOP1) inhibitors that are widely used to treat cancer. Topotecan was shown to bind to U4 snRNA, competing with SNU13, and to decrease RNA splicing efficiency on a splicing reporter construct [342]. Thus, presumably, topotecan interfered with U4/U6 di-snRNP formation. However, while TOP1 inhibitors were developed to target TOP1 DNA-relaxing activity, TOP1 has also been shown to play a role as a kinase regulating AS. Although TOP1 lacks a canonical ATP-binding site, it was shown to be capable of phosphorylating several SR proteins including SRSF1. Treatment of cells with topotecan reduced SR protein phosphorylation [343]. The TOP1 inhibitor NB-506 reduced TOP1-dependent phosphorylation of SRSF1, but not phosphorylation by other SR protein regulating kinases SRPK1 or CLK. In HeLa nuclear extracts, NB-506 prevented early spliceosome assembly and inhibited splicing of a β-globin pre-mRNA substrate. In mouse leukemia cells, but not in the same cells with mutations in the TOP1 gene that confer resistance to inhibitors, NB-506 decreased the expression of the anti-apoptotic Bcl-xL isoform and CD44 isoforms associated with metastatic behavior [344]. These observations spurred investigations to dissect the TOP1 DNA-relaxing and kinase activities and discovery of specific TOP1 kinase inhibitors [345]. Structural information on the interaction between TOP1 and SRSF1 aids this development [346]. While specific inhibitors for the regulatory effect of TOP1 on mRNA splicing have to date not been discovered, it is possible that the anticancer effects of approved TOP1 inhibitors such as topotecan and irinotecan are in part caused by changing AS. It might be difficult, however, to distinguish between the DNA-relaxing and mRNA splicing regulating effects of TOP1 inhibitors on clinical benefit.

The aim of another study was to interfere with conformational changes in the late spliceosome required to form the active site for the splicing reaction. Here, the NTC core components CDC5L and PLGR1 act in direct association. In an attempt to inhibit mRNA splicing by interfering with their protein–protein interaction (PPI), peptides derived from their interaction surfaces were designed [347]. Several of these peptides bound to their partner protein, without preventing the CDC5L:PLGR1 PPI. Spliceosome assembly was not inhibited, but splicing of several pre-mRNA substrates was. Hence, apparently the peptides interfered with PPI contacts between CDC5L and PLGR1 that are essential for mRNA splicing [347]. The RNA-binding protein KHDC4 that affects both 5′ and 3′ splice site selection is also part of the NTC, where it associates with PRPF19 [348]. Using a Systematic Evolution of Ligands by EXponential enrichment (SELEX) approach with recombinant KHDC4 or its conserved RNA binding domain as bait, an RNA-aptamer was identified that specifically binds to KHDC4 and inhibits splicing in vitro, as demonstrated by the accumulation of synthetic pre-mRNA and a slight inhibition of spliceosome assembly [349]. Presumably, the aptamer inhibits binding of KHDC4 to its target sequence on the pre-mRNA, thereby preventing proper positioning of the NTC and activation of the catalytic site. While the peptides and aptamers targeting NTC core components were developed primarily as tools to study the functions of the interacting NTC proteins in the mRNA splicing reaction, they could perhaps also serve to develop (peptidomimetic) drugs.

#### 4.2.7. Discovery of Inhibitors of Spliceosome Catalytic Activity

Studies were also undertaken to develop inhibitors of mRNA splicing factors that regulate AS by interfering with the catalytic step. Although RBM17 (SPF45) is perhaps already recruited in an early stage of spliceosome assembly, it regulates 3′ splice site utilization during the second catalytic step [350]. For this activity, an UHM domain in RBM17 is essential. Presumably, RBM17 interacts via this motif with multiple ULMs in constitutive spliceosome components such as SF3B1 or U2AF65 [351]. RBM17 was shown highly expressed in many cancer types and experimental overexpression was associated with drug resistance [352]. Silencing RBM17 induced the pro-apoptotic exon 6 inclusion splicing switch in the FAS gene [351]; as well as cell cycle arrest and apoptosis in glioblastoma cells in vitro and decreased tumor growth in a glioblastoma xenograft model in vivo [353], which makes it an interesting target for treatment. On the basis of the crystal structure of synthetic RBM17 UHM-SF3B1 ULM complexes and peptide optimization studies, a cyclic peptide was designed that binds to the RBM17 UHM domain with 270-fold selectivity compared to the U2AF65 UHM. The peptide inhibited mRNA splicing and early spliceosome assembly [354]. This suggests that early recruitment of RBM17 is essential for spliceosome assembly, or, alternatively, the designed peptide inhibited other UHM-ULM interactions that are essential for early spliceosome assembly. The latter would not be entirely unexpected, since the peptide was derived from one of the SF3B1 UHMs and might thus also compete with SF3B1 for binding to other interaction partners. The cyclic peptide was subsequently used to discover small molecules that target the UHM-ULM interaction. To this end, the peptide was fluorescently labeled and used with synthetic RBM17 UHM in a fluorescence polarization assay, testing competition by over 43,000 compounds [355]. A validated hit was used for SAR and structure based optimization studies. The identified phenothiazine compounds appeared general inhibitors of UHM-ULM interactions that impair early spliceosome assembly. While most compounds inhibited complex A formation, one analogue inhibited transition to complex B, presumably by preventing docking of the U4/U6.U5-tri-snRNP [355]. Although this difference could perhaps be exploited to design more specific inhibitors, the high conservation of UHM-ULM interactions between various proteins in the spliceosome makes this a challenging task. Another mRNA splicing factor with an UHM domain is RBM39 (CAPERα). RBM39 has a generally opposite effect on AS compared to RBM17, with RBM39 promoting exon inclusion and RBM17 reducing exon inclusion [356]. RBM39 overexpression appears associated with oncogenesis and knockdown or knockout of RBM39 reduced the viability of CRC and AML cells [357,358,359]. Via its UHM domain, RBM39 associates with U2AF65, perhaps only in the presence of U2AF35 [360,361,362]. RBM39 also binds to ULMs of SF3B1 [363], but it is not known if these proteins interact directly in living cells. In a target identification study to resolve the mechanism of action of anticancer sulfonamides, Uehara et al. [364] found that the sulfonamides indisulam, E7820 and CQS promoted proteasomal degradation of RBM39 by inducing its complexation with the ubiquitin ligase adapter protein DCAF15. Importantly, there was a high correlation in the transcriptomic changes induced by E7820 treatment and RBM39 silencing in cancer cells [364]. In another study, treatment with indisulam inhibited DNA synthesis and induced apoptosis in AML cells in vitro; and reduced leukemic burden and extended survival in mouse AML cell line and patient-derived xenograft models [359]. As expected in view of its mechanism of action, indisulam sensitivity was dependent on DCAF15 expression. Interestingly, AML cells carrying mutations in spliceosome genes were more sensitive to treatment with sulfonamides than wild type cells. Treatment of AML cells with E7820 induced more profound AS events, in particular exon skipping and intron retention events, than treatment with the SF3B1 inhibitor E7107 [359]. E2820 is an orally bioavailable drug that is currently in development through multiple clinical trials.

#### 4.2.8. Discovery of Inhibitors of mRNA Splicing Factors and Regulatory Kinases

Many attempts were made to inhibit mRNA splicing factors or kinases that regulate their activity. Kinases are particularly attractive targets because they are considered highly druggable. However, many of the kinases that regulate the activity of mRNA splicing factors have multiple functions. Therefore, the anticancer efficacy of candidate drugs that inhibit these proteins cannot always be directly attributed to inhibition of mRNA splicing. Here, we discuss primarily molecules for which modulatory effects on mRNA splicing were shown.

Not many drugs were designed to target SR proteins or hnRNPs. A structure-based drug discovery approach was taken to identify small molecules that could bind to the RNA-binding pocket of hnRNPA1, to interfere with pre-mRNA binding and AS modulation [365]. Virtual screen hits were experimentally validated using a reporter assay based on AS of the androgen receptor into the ARv7 variant, which is associated with castration-resistant prostate cancer, conferring drug resistance; and is known to be promoted by hnRNPA1. This identified the compound VPC-80051 that interacted directly with the RNA-binding domain of hnRNPA1 and reduced the ARv7 variant in androgen-deprived castration-resistant prostate cancer cells [365]. Further drug optimization and biological validation could offer opportunities to develop cancer treatments with hnRNPA1 inhibitors. Another category of spliceosome components with enzymatic activity that could potentially be inhibited with drugs are the RNA helicases, which are important to catalyze conformational changes in the dynamic spliceosome during the splicing process. The Brr2/SNRNP200 helicase is involved in the ATP-dependent unwinding of the U4/U6 snRNA duplex, representing a crucial step for spliceosome activation. Since screening for helicase activity is challenging, high-throughput compound screening for SNRNP200 inhibitors was done with RNA-dependent ATPase activity as readout. Through SAR probing, co-crystallization studies and chemical optimization by structure-based drug design, molecules with selective SNRNP200 ATPase and helicase inhibitory activity were identified [366,367]. It remains to be determined what the biological and therapeutic relevance of these compounds are.

As discussed in Section 2.2.1, the activity of SR proteins is regulated by phosphorylation. In particular, SR protein kinases 1 and 2 (SRPK1/2) and several CLK and DYRK family members are known to phosphorylate SR proteins, which affects their alternative splicing regulatory activity. These kinases are often dysregulated in cancer and thus represent putative targets for treatment. Therefore, there has been a strong incentive to develop inhibitors of these kinases.

Several SRPK1/2 inhibitors were developed and tested for their effects on mRNA splicing. The ATP-competitive SRPK1/2 inhibitor SRPIN340, a trifluoromethyl arylamide that was originally identified by screening a chemical library for specific SRPK inhibitors in search for antiviral agents [368], reduced phosphorylation of all tested SRSFs in leukemic cell lines and displayed cytotoxic effects on these cells in vitro. Treatment with SRPIN340 reduced expression of the pro-angiogenic VEGF165 isoform and increased expression of the pro-apoptotic FAS isoform [369]. In a melanoma xenograft model, treatment with SRPIN340 reduced tumor cell growth [370]. A compound designed based on the structure of SRPIN340 with an aryl bromide group exhibited more potent activities [371] and another derivative, SPHINX, exhibited more selectivity for SRPK1 [372]. Rational design based on a high resolution crystal structure of SRPK1 with a SPHINX analogue led to development of SPHINX31 [373]. When tested for binding to SRPK, CLK and DYRK family members, it exhibited a 50–100-fold higher affinity for SRPK1 than for SRPK2, CLK1 and CLK4; and negligible binding to the other tested kinases. In an ATP competition assay against 50 kinases, SPHINX31 did not exhibit detectable inhibition of kinases other than SRPK1. In cells, SPHINX31 inhibited SRSF1 phosphorylation and splicing of the VEGF165 gene to produce the anti-angiogenic variant [373]. The irreversible SRPK1/2 inhibitor SRPKIN-1 was developed from the FDA-approved ALK-inhibitor Alectinib. By searching profiling data of known kinase inhibitors, Alectinib was found to bind SRPK1. Structural analysis of Alectinib bound to SRPK1 and SAR probing identified a more active compound. From this compound, SRPKIN-1 was derived to comprise a sulphonyl fluoride substituent that can form a covalent bond with a unique tyrosine located immediately adjacent to the ATP-binding pocket of SRPK1 [374]. SRPKIN-1 had high activity against SRPK1 and SRPK2, but not against ALK or any other kinase in a large panel tested. SRPKIN-1 inhibited phosphorylation of all SRSFs analyzed and induced the anti-angiogenic splicing switch in VEGF. In a murine retinal model, this inhibitor was capable of blocking neovascularization. In all these analyses, SRPKIN-1 outperformed SRPIN340 [374].

Recognizing that CLKs influence AS by regulating SR activity [375], several CLK inhibitors in addition to those that were identified in functional screens were investigated as indirect modulators of mRNA splicing to treat cancer. By screening 100,000 chemical compounds for in vitro phosphorylation by CLK1, the inhibitor TG003 was identified. TG003 inhibited CLK1 and CLK4, and to a lesser extend CLK2; reduced phosphorylation of SRSF1 and changed AS patterns [376]. Because TG003 exhibited cross reactivity with several other kinases, attempts were made to develop more specific CLK1/4 inhibitors. Serendipitously, a side product in the synthesis of a derivative of a natural alkoid with anti-proliferative activity was found to potently inhibit CLK1 and CLK4 when tested against a panel of kinases [377]. In stimulated endothelial cells with increased SR phosphorylation, this compound KH-CB19 reduced this phosphorylation much more potently than did TG003 [377]. When mouse embryo stem cells were treated with KH-CB19 and analyzed for detained introns, many splicing changes were observed, including reduced intron detention in the CLK1 and CLK4 genes and mRNA splicing changes in several SR genes and many other mRNA processing genes. Analysis of SR protein phosphorylation revealed that the predominant effect of KH-CB19 was inhibition of SRSF4 phosphorylation, suggesting that SRSF4 is an important regulatory target for CLK1/4 [378]. The potent CLK/DYRK dual inhibitor T-025 was found to induce alternative splicing in breast cancer cells in vitro and in xenograft tumors in vivo; and to inhibit cancer cell proliferation in vitro and tumor growth in vivo. Cancer cell lines containing MYC amplification were especially sensitive to T-025 treatment, making this a possible biomarker for CLK inhibitor treatment stratification. Moreover, the efficacy of T-025 treatment correlated with CLK2 expression and with the degree of AS that was induced. Hence, T-025 modulated AS in cancer cells, culminating in the induction of cell death, with a magnitude depending on its target CLK expression [379]. Another kinase inhibitor, CC-671, targets CLK2 and dual specificity protein kinase TTK. It was developed by optimizing a hit molecule from a synthetic lethality screen on TNBC cells versus luminal breast cells [380]. While CC-671 inhibited phosphorylation of SRSF4 and SRSF6 and induced AS of many genes in TNBC xenograft tumors in vivo [381], it is not clear if the observed antitumor effects of CC-671 in preclinical TNBC models [380,381] that supported its subsequent clinical development are due to modulation of mRNA splicing. By screening an 870,000-compound library for inhibitors of SRPK1 and CLK2, three dual activity inhibitors were identified, two of which were particularly potent against CLK1 and CLK2. These compounds induced SRSF1-dependent AS by inhibiting CLK activity in cancer cells [382]. By chemical modification of the structure of these compounds the inhibitor T3 was designed, which exhibited high specificity for the CLK1-3 proteins. In a head-to-head comparison with KH-CB19, T3 exhibited strong inhibition of SRSF1, SRSF4 and SRSF6 phosphorylation and increased cytotoxicity [383]. T3 decreased exon recognition consistent with SR protein inhibition. The majority of AS events induced by T3 were common between different cell types. In addition, T3 increased the number of transcripts originating from so-called conjoined genes, i.e., transcripts formed by splicing between two adjacent genes. This unanticipated observation was probably associated with aberrant 3′-end processing caused by inhibition of CLKs regulating SR proteins in complexes with 3′-end processing factors [383]. The compound SM08502 was picked up in a screening campaign for CLK inhibitors and found to be most potent against CLK 2 and CLK3. The latter distinguishes this compound from other small molecules targeting CLKs [384]. SM08502 potently inhibited SRSF5/6 phosphorylation and altered splicing of Wnt pathway genes in CRC cells. The latter appeared to require combined inhibition of CLK2 and CLK3. Expression of several Wnt-pathway genes was reduced, presumably via the formation of unstable transcript variants. SM08502 reduced viability of gastrointestinal cancer cells in vitro and inhibited growth of xenograft tumors in vivo; and is currently in clinical development [384]. While several other CLK inhibitors with various specificity profiles have been described and shown to have anticancer activities, these have not always been associated with modulation of mRNA splicing.

#### 4.2.9. Discovery of Inhibitors of Sm-snRNP Core Complex Assembly

Finally, there have also been studies aiming at inhibition of Sm-snRNA core complex production in the cytoplasm. This way, the pool of U snRNP units in the nucleus is depleted and mRNA splicing impeded. Sm-snRNA core complex construction requires the formation of the ring-shaped 7 Sm protein structure. Since for three of the seven Sm proteins symmetrical dimethylation of C-terminal arginine residues by PRMT5/7 is essential for their interaction with SMN during Sm ring assembly, inhibition of these enzymes offers an opportunity to block Sm-snRNA core and thus subsequent U snRNP production. PRMT5 and the Sm genes are overexpressed in lung cancer and their silencing induced AS, activated p53 presumably via AS of MDM4, inhibited cell proliferation and induced cell death [268,273,274,385,386]. Thus, inhibition of Sm-snRNA core complex production could have therapeutic utility.

Most studies focus on inhibition of PRMT5 activity, not only to inhibit mRNA splicing, but also to inhibit other functions of PRMT5, including histone methylation. Structure based design of PRMT5 inhibitors resulted in the synthesis of DS-437 [387]. When profiled against a panel of human methyltransferases, DS-437 exhibited clear activity against PRMT5 and PRMT7, with residual activity against DNMT3A/B. In breast cancer cells, DS-437 inhibited dimethylation of SmB and SmD1/3 [387]. So far, effects of DS-437 on AS have not been investigated. The orally bioavailable PRMT5 inhibitor EPZ015666 was discovered by screening a library containing 370,000 small molecules for inhibition of methylation of a histone peptide by recombinant PRMT5 in complex with its co-factor WDR77 (MEP50), followed by SAR exploration and structure-based optimization [388,389]. EPZ015666 was highly selective for PRMT5 against other protein methyltransferases; exhibited dose-dependent inhibition of SmD3 dimethylation in mantle cell lymphoma (MCL) cells; and inhibited MCL cell proliferation in vitro and MCL xenograft tumor growth in vivo [388]. Another compound from the same series of analogues, GSK3326595, is currently in clinical trials for multiple cancers (NCT02783300; NCT03614728).

In an ELISA-based high-throughput screen, 10,000 compounds were tested for inhibition of methyl transfer to SmD3 by recombinant PRMT5-WDR77 complex. This identified 8 compounds that were not structurally related to known PRMT5 inhibitors [390]. The compounds inhibited methylation of SmD3 and histones by PRMT5, but not by PRMT1/3, impeding NSCLC cell growth in vitro. A more water soluble and orally bioavailable analogue of one of the compounds decreased tumor growth in a lung cancer xenograft mouse model. The compound was not without toxicity, as it also retarded growth of the mice. It was not investigated which inhibited PRMT5 target protein methylation accounted for the antitumor effect [390]. In another high-throughput compound screen analyzing for inhibition of methylating activity of recombinant PRMT5 on a histone peptide, the structurally nonrelated molecules P1608K04 and P1618J22 were identified. These compounds exhibited a cytotoxicity on pancreatic and colorectal cancer cells with IC50 values well below that of EPZ015666. Selectivity for PRMT5 and inhibition of mRNA splicing were not investigated [391].

Interestingly, when a panel of 45 drugs targeting spliceosome components and epigenetic regulatory proteins was tested in a cell viability screen on an isogenic pair of AML cell lines with wild type and mutant SRSF2, the SRSF2 mutant cells were particularly sensitive to the SF3B1 inhibitor E7107, the PRMT5 inhibitor EPZ015866 and the PRMT1/3/4/6 inhibitor MS023 [392]. EPZ015866 is a more potent analogue of EPZ015666 that was considered less suitable for in vivo use [389]. Dose-response experiments showed that SRSF2 mutant cells were approximately 10-fold more sensitive to either PRMT inhibitor than SRSF2 wild type cells, which was mirrored by differences in inhibition of methyltransferase activities in these cells. This finding was validated on mononuclear cells from AML patients, where EPZ015866 was more effective against cells with SRSF2, U2AF1, or SF3B1 mutations. Also, in a mouse leukemia model in vivo, PRMT5 inhibition with EPZ015866 was more effective in the context of mutant SRSF2; and this effect was synergistic in combination with MS023 treatment. Synergy between all combinations of the three drugs was further seen in human SRSF2 or SF3B1 mutant AML cell lines and primary samples as well as PDX models [392]. Proteomics analysis for methylated peptides revealed that EPZ015866 and MS023 treatment induced distinct methylation changes, mainly in RBPs involved in RNA processing including SmB and hnRNPA1 involved in mRNA splicing. The two drugs induced AS in SRSF2 wild type and mutant cells, with combination treatment inducing a unique pattern. Combination treatment AS effects in SRSF2 mutant cells showed enriched perturbations in cell cycle regulation and DNA repair genes. Among these AS events, restoration of mutant SRSF2-driven aberrant splicing of EZH2 could be shown to contribute to the effect of combined EPZ015866/MS023 treatment [392].

The PRMT5 co-factor WDR77 is phosphorylated by cyclin-dependent kinase 4 (CDK4)-cyclin D complexes. Several CDK4/6 inhibitors are in use to treat a variety of cancers with hyperactive CDK4/6-cyclin D1. Treatment of melanoma cells with one of these inhibitors, palbociclib, decreased PRMT5 activity in sensitive cell lines, reducing target protein methylation and inducing the MDM4 AS switch known to be associated with PRMT5 inhibition. Subsequent p53 activation, p21 upregulation and cell cycle arrest was a major response to palbociclib treatment in these cells [393]. Palbociclib resistant cell lines were sensitive to the PRMT5 inhibitor GSK3326595; and combination treatment with palbociclib and GSK3326595 caused a more permanent cell cycle inhibition than single agent treatment, also in palbociclib-resistant cells. These effects were dependent on MDM4 expression, suggesting that the AS of this gene was critical. However, the palbociclib-GSK3326595 combination effect was also seen in p53 deficient cells, suggesting that the PRMT5-MDM4-p53-p21 axis is only one of the mechanisms of GSK3326595 enhancing the efficacy of palbociclib. Treatment of melanoma xenografts with palbociclib in combination with GSK3326595 led to a prolonged tumor growth control. Administration of GSK3326595 when resistance to single-agent palbociclib treatment emerged also delayed tumor regrowth [393]. Thus, combination treatment with CDK4/6 and PRMT5 inhibitors, which are already in clinical use, could be considered for tumors with high MDM4 expression.

Another strategy to interfere with Sm-snRNA assembly could be to inhibit crucial PPIs. Rather than prohibiting the symmetrical dimethylation of Sm proteins by PRMT5/7, the interaction of symmetrical dimethylated Sm protein with SMN could be impeded. Symmetrical dimethylation of SmD1, SmD3, SmB/B’ and LSm4 occurs at C-terminal GRG motifs in these proteins. A synthetic peptide containing nine symmetrical dimethylated RG dipeptides was shown to compete with Sm proteins and U snRNPs for binding to SMN [87]. To develop this finding into a therapeutic peptide, a polypeptide was designed that contains an N-terminal cell-penetrating peptide derived from protegrin and a C-terminal (GR)6 peptide, connected via an elastin-like polypeptide that aggregates above a certain transition temperature and is soluble below [394]. The proposed use of the polypeptide is to administer it systemically and to allow it to aggregate and accumulate at a tumor site by applying local hyperthermia. The peptide accumulated in the cytoplasm of cultured cancer cells when temperature was increased to 42 °C. This inhibited the interaction between SMN and SmB and induced apoptosis [394]. It is unclear if the cytotoxicity was associated with impaired spliceosome function. The peptide formed aggregates in the cytoplasm, which in itself could be toxic. Furthermore, peptides are generally less suitable for drug development than small molecules. Nevertheless, the SMN-Sm PPI activity of symmetrical dimethylated GRG-peptides could perhaps be a starting point for development of a peptidomimetic drug. Furthermore, it suggests that PPI of core spliceosome proteins could be a viable therapeutic option.

## 5. Concluding Remarks and Future Perspectives

To maximize efficacy and minimize treatment-related side effects, targeted treatments of cancer are actively being sought. Many of these treatments capitalize on the acquired biological capabilities that are considered hallmarks of cancer [395]. The success of targeted treatments relies on effectively interfering with truly cancer-selective hallmarks, as this should have little effect on non-malignant cells. However, resistance to treatment might emerge through cancer cells adapting to or evading loss of a targeted hallmark capability. Inhibiting multiple cancer-specific vulnerabilities—to prevent or to counteract emerging compensatory salvage pathways—could result in a more effective treatment of cancer. The proposed 11th hallmark of epigenetic and RNA dysregulation in cancer, with mutant genes found in 22% of NSCLC cases [191], provides additional opportunities for targeted treatment. The processes that are dysregulated in this hallmark of cancer are largely coordinated in the supraspliceosome. Here, in particular, the mRNA splicing machinery appears to offer promising targets for cancer treatment.

In lung cancer, specific changes in the mRNA splicing pattern of individual genes have been detected. While these could perhaps be exploited as targets for treatment, this is not likely to be very effective. AS can be influenced by the use of splice-switching antisense oligonucleotides, but efficient delivery of these molecules to cancer cells in the human body is a major challenge [396]. In addition, distinct mRNA splicing changes were found in only a minority of NSCLC patients. Hence, specifically targeting these splicing aberrations will have relatively narrow applicability. Most splicing aberrancies in lung cancer are caused by dysregulations in the mRNA splicing machinery, affecting the AS of many genes in parallel. Thus, in theory, targeting the dysregulated spliceosome is a more effective way of treating cancer. This consideration has motivated a plethora of investigations into the identification of targets in the spliceosome and their specific pharmacological inhibitors. As can be expected for a huge mRNA processing machinery with complex and dynamic protein–protein and protein–mRNA interactions, many vulnerabilities were identified in the spliceosome. Gene silencing experiments validated the role of identified targets in maintaining aberrant AS patterns and the associated biological characteristics in cancer cells. Many small molecules were found to inhibit the spliceosome, to change AS in cancer cells and to exhibit anticancer properties. In addition, several molecules that were already in use as anticancer agents were found to affect mRNA splicing, suggesting that this property could contribute to their therapeutic efficacy. However, not for all inhibitors the exact target in the spliceosome has been recognized; and several molecules target kinases that activate regulatory splicing factors, but that also have activities with functional consequences distinct from mRNA processing. Hence, there is not always formal proof that the anticancer efficacy of mRNA splicing inhibitors can be attributed to inhibition of the spliceosome. Nevertheless, these studies have delivered a wealth of new leads for the development of anticancer medicines. Which of the many strategies to target the mRNA splicing machinery is the most promising to treat cancer is difficult to say. Side-by-side comparisons of different spliceosome-targeting drugs in preclinical models have not been made and only few experimental drugs have reached clinical investigation. However, with many drug development programs on the way, more insight into the utility of different inhibitors of the mRNA splicing machinery will be gained in the coming years.

An important observation has been that inhibition of core components of the spliceosome is a particularly selective vulnerability of cancer cells. While complete loss of these proteins is expectedly incompatible with cell viability, a certain degree of inhibition appears to be tolerated by non-malignant cells. In contrast, cancer cells are effectively killed upon their genetic or pharmacological suppression. For several core components of the spliceosome, mutations were found associated with cancer, constituting the proposed hallmark of cancer. However, several other spliceosome proteins that appeared particularly useful targets are only rarely mutated in cancer. This includes, e.g., the Sm proteins that we identified as selective lethal targets in NSCLC cells. They were generally overexpressed and regularly amplified, but almost never mutated in NSCLC tumors [274]. Apparently, cancer cells rely heavily on increased mRNA splicing activity to maintain their dysregulated mRNA splicing programs. Therefore, overexpression of spliceosome components might perhaps be considered a non-oncogene addiction required to overcome cancer-associated cellular stresses [397]. Further investigations into this vulnerability and discovery of selective inhibitor molecules could lay the foundation for the development of new treatments for NSCLC, to contribute to combatting this devastating disease.

## Figures and Tables

**Figure 1 ijms-22-05110-f001:**
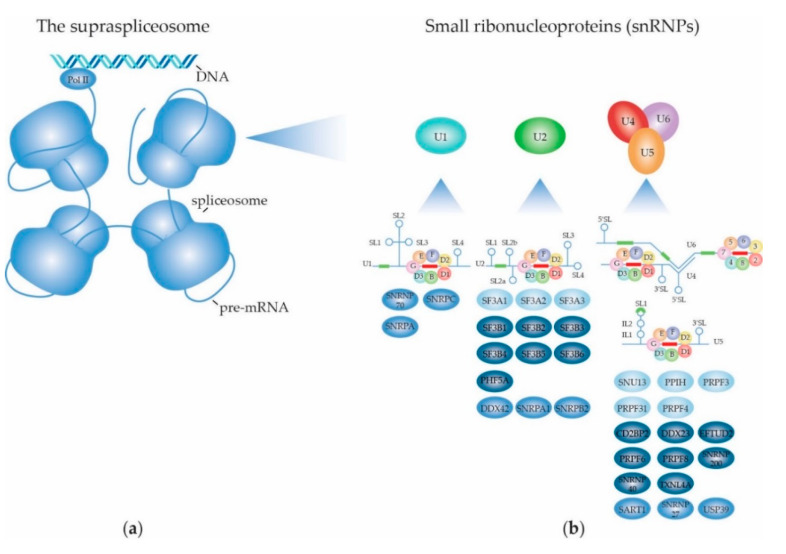
Structure and composition of the supraspliceosome. (**a**) Schematic impression of the macromolecular structure of the supraspliceosome, consisting of four native spliceosomes assembled onto a pre-mRNA being transcribed by RNA Pol II. In addition to pre-mRNA splicing carried out by the spliceosomes, the supraspliceosome provides a platform for all other co-transcriptional mRNA processing activities (by proteins that are not shown in the illustration). (**b**) Major components of the spliceosome U1, U2, and U4/U6.U5 snRNP subunits, with their Sm-snRNA core complexes and subunit-specific splicing factor proteins (U2: SF3a complex in light blue, SF3b complex in dark blue; U4/U6.U5: U4/U6-specific proteins in light blue, U5-specific proteins in dark blue). SL, snRNA stem-loop; red box, snRNA Sm site; green box, other RNA-binding domains (U1: 5′ss recognition sequence, U2: BPS recognition sequence, U6: U2/U6 duplex-forming sequence).

**Figure 2 ijms-22-05110-f002:**
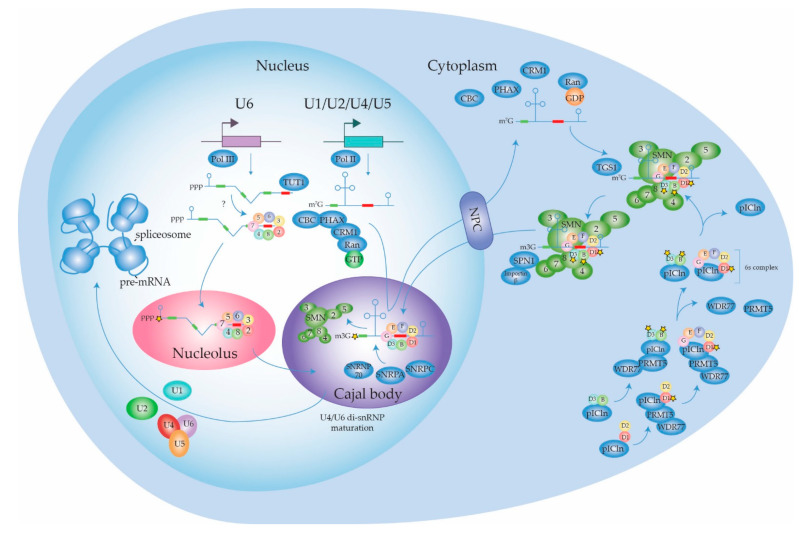
Spliceosome snRNP subunit biogenesis. Details are given in the main text. Since U1, U2, U4 and U5 snRNPs are compiled in the same manner, the biogenesis of U1 snRNP is shown as example. The snRNAs are transcribed by either RNA polymerase II (U1, U2, U4, U5) or RNA polymerase III (U6). RNA polymerase II-transcribed snRNA is exported to the cytoplasm after quality control in Cajal bodies (CBs). SmD1, SmD3 and SmB/B’ are dimethylated (displayed by stars) in the cytoplasm by the methylosome complex. The Sm ring is assembled around the snRNA by the SMN complex and the Sm-snRNA core complex is imported back into the nucleus where snRNP maturation takes place in the CBs. The exact LSm assembly route onto the U6 snRNA is not known but presumably takes place in the nucleus. The LSm-U6 snRNA core complex probably moves to the nucleolus where it is 2-O’-methylated (displayed as a star) and subsequently transported to the CBs where U6-specific proteins are assembled and U4/U6.U5 tri-snRNP maturation can take place. Mature snRNPs are recruited into the supraspliceosome to take part in the RNA splicing reaction.

**Figure 3 ijms-22-05110-f003:**
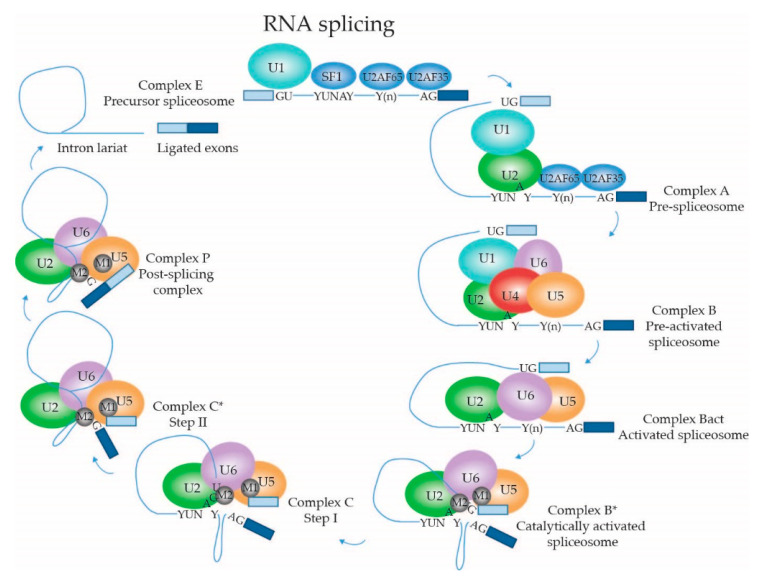
Pre-mRNA splicing reaction performed by the spliceosome. The dynamic composition of the spliceosome, with it different intermediate complexes, is illustrated. Details are given in the main text. Light and dark blue boxes, exons; line, intron; GU, 5′ splice site; AG, 3′ splice site; YUNAY, branchpoint sequence; Y(n), polypyrimidine tract; M1 and M2, Mg^2+^ metal ions at the catalytic site.

**Figure 4 ijms-22-05110-f004:**
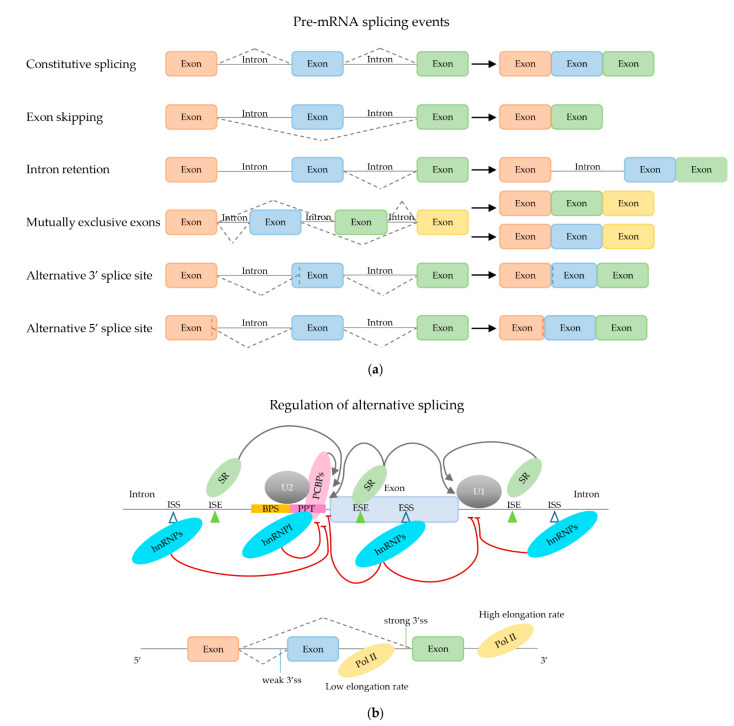
Pre-mRNA splicing events and regulation of alternative splicing. (**a**) Schematic representation of constitutive and common types of alternative splicing events. (**b**) Pre-mRNA splicing regulation. Upper figure: Alternative splicing is mainly regulated by trans-acting splicing factors (such as hnRNPs and SR proteins, which usually inhibit and promote splice site usage, respectively) binding to cis-acting splicing regulatory elements (ESE, ISE, ESS and ISS) on the pre-mRNA substrate. Lower figure: alternative splicing also depends on the elongation rate of RNA polymerase II (Pol II). A low elongation rate of RNA Pol II might provide more opportunities to weak 3′ss usage; while a high elongation rate of RNA Pol II might give priority to strong 3′ss usage. Open triangle: splicing silencer element; filled triangle: splicing enhancer element; arrow: promoting; –I: inhibiting.

**Figure 5 ijms-22-05110-f005:**
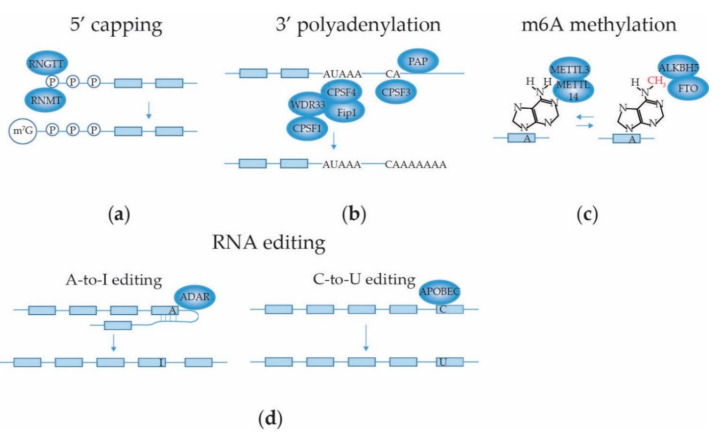
Illustration of other (pre-)mRNA processing activities performed in the supraspliceosome simultaneous with pre-mRNA splicing. (**a**) 5′ capping, (**b**) 3′ polyadenylation, (**c**) m6A methylation, (**d**) A-to-I and C-to-U RNA base editing. Details are given in the main text.

**Figure 6 ijms-22-05110-f006:**
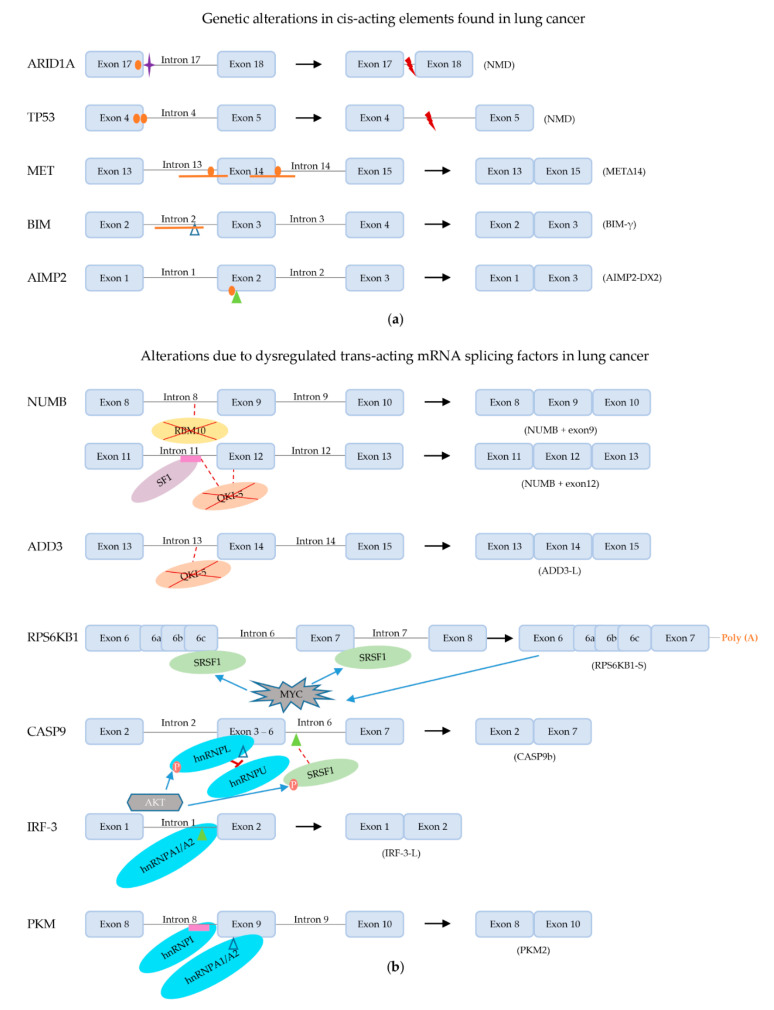
Dysregulation of mRNA splicing in lung cancer (**a**) Genetic alterations in cis-acting elements found in lung cancer. The figure shows aberrant splicing outcomes of lung cancer-related genes with mutations in splicing sites (ARID1A, TP53, MET) or cis-acting regulatory elements (BIM, AIMP2). (**b**) Alterations due to dysregulated trans-acting mRNA splicing factors in lung cancer. Aberrant splicing outcomes of lung cancer-related genes NUMB, ADD3, RPS6KB1, CASP9, IRF-3, and PKM due to splicing factor mutation or abnormal expression. Circle: point mutation; line: deletion mutation; star: intronic cryptic site activation; lightning bolt: premature termination codon in retained intron; open triangle: splicing silencer element; filled triangle: splicing enhancer element; filled rectangle: polypyrimidine tract; P: phosphorylation; dashed line: unbound; arrow: promoting; –I: inhibiting; crossed-out splicing factor: mutational loss.

**Figure 7 ijms-22-05110-f007:**
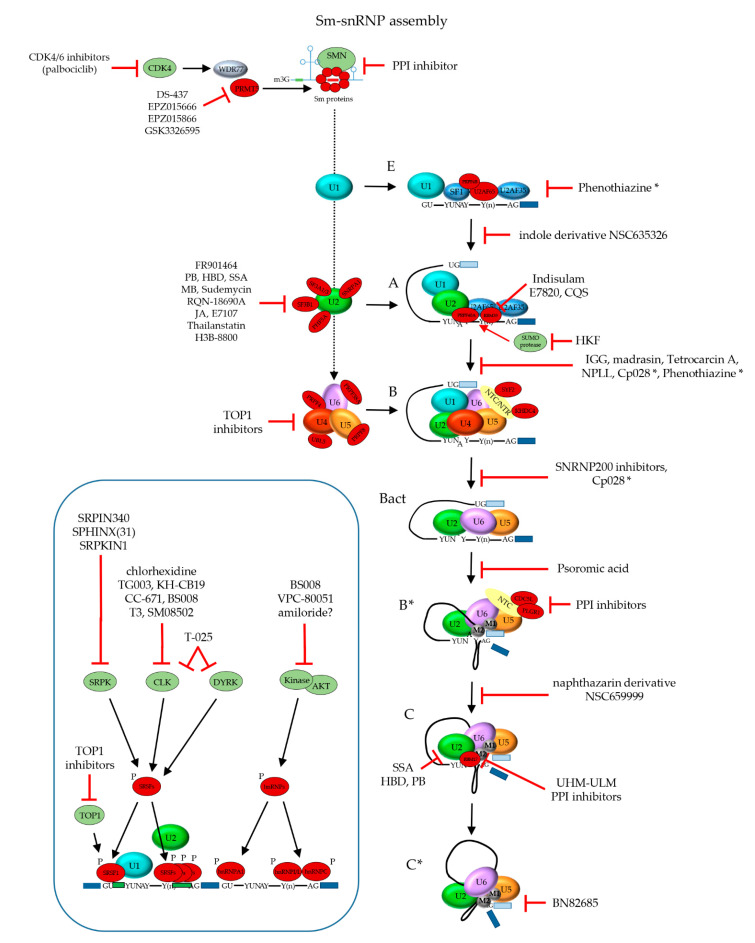
Therapeutic targets in the spliceosome and inhibitors of the mRNA splicing reaction. Main figure: structural components of the spliceosome needed for the mRNA splicing reaction; insert: splicing factors that regulate splice site usage. Identified targets in the spliceosome are shown in red. Main representative molecules and classes of splicing inhibitors are shown, with an indication of their interference with the process. For several inhibitors, the target protein is known; for others only the spliceosome complex that is inhibited, or the complex transition step that is prevented. In cases where there is doubt about the step that is inhibited, the compound is shown at multiple steps and marked with an asterisk.

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
