# Peer review of "Biology of the mRNA Splicing Machinery and Its Dysregulation in Cancer Providing Therapeutic Opportunities"

_ijms, 2021, doi:10.3390/ijms22105110_

Round 1

Reviewer 1 Report

This ambitious review reports the state of the art of knowledge of the splicing process, ranging from the basis of splicing to the connection with other mRNA processing cellular activities. Moreover, they report current knowledge of splicing dysregulation in lung cancer as well as in other cancer cells. In addition, they report investigations to discover new therapeutic targets as well as an overview of inhibitors and modulators of the mRNA splicing process identified so far.

This review is very informative but, even considering its comprehensive scope, it is rather on the long side and could be considerably streamlined. Moreover, it is important to analyze, rather than solely describe the data in terms of the underlying mechanisms but also the efficacy obtained in vitro and in vivo models. This will guide the reader to the best, so far, strategy.

The structure and presentation of the material could be improved as follows:

  • The review is rather on the long side and could be considerably streamlined. In particular, the first part of the review gives a very detailed overview of splicing and its connection with other cellular activities. In fact, by excluding the pages for References, 19 out of 45 pages are dedicated to the description of splicing mechanisms. Therefore, I suggest shortening the first section. Alternatively, authors could split the review into two reviews, the first one reporting the current findings on splicing and its connection with other cellular activities, and the other one focused on alteration of splicing in the lung cancer model.
  • In light of point 1, I think the title of the review could be improved to reflect the general contents of the manuscript.
  • In the review, in different parts, it is reported that the spliceosome builds up around the intron, so giving the misconception that spliceosome is only able to define the intron. However, there are several experimental evidences that spliceosome can build up also around the exon, in particular when introns are very long. So, authors should include also this concept in the review.
  • On page 12, line 523, the authors state that “the strength of a splice site is dependent on the gene context”. Well, while this is true when only considering the exon-intron sequence, it is well known that the “strength” of splice sites, as also reported by authors, is influenced by many factors, including polymerase speed, nucleosome position. Therefore, the authors should smooth this sentence.
  • In light of point 4, the authors are forgiving to mention that also the RNA secondary structure can influence the splicing outcome, by influencing splice site recognition or binding of splicing factors.
  • Authors should also include, among other splicing affecting factors, also the so call istonic code, where a different splicing outcome is observed in presence of different istonic protein modification, thus relating the splicing with chromatin structure and epigenetic features. Moreover, authors should also include the importance of exonization events in presence of Ale elements, an important feature for the creation of new exons and thus the evolution of complexity.
  • In the last section of the review, the authors report an overview of investigations to discover therapeutic targets in the spliceosome and give an overview of 18 inhibitors and modulators of the mRNA splicing process identified so far. For sake of completeness, authors should also report AON (ESSENCE, TOE) and U1snRNA based strategies for the modulation of splicing outcome, with therapeutic potential also for cancer cells.
  • I suggest including more subheadings, to help readers in following the review.

Author Response

Moreover, it is important to analyze, rather than solely describe the data in terms of the underlying mechanisms but also the efficacy obtained in vitro and in vivo models. This will guide the reader to the best, so far, strategy.

Response: This remark relates to section 4 where we discuss therapeutic strategies. In section 4.1, where we discuss therapeutic target discovery studies that were mainly done using RNAi, we everywhere already mention the therapeutic effects observed upon gene silencing, such as e.g. induction of apoptosis or inhibition of resistance to chemotherapy. In these studies, therapeutic effects were usually evident. In section 4.2, where we discuss new drug development, however, such data are not always available or therapeutic efficacy is less evident. While some molecules were discovered in synthetic lethality screens on cancer cells and their efficacy in killing cancer cells is thus obvious, other molecules were designed to target splice factors and are only assumed to have anticancer efficacy based on the known effects of silencing their targets. Many drug discovery programs are still in the phase of lead molecule discovery or optimization of target inhibition. For the molecules where we identified data on treatment efficacy in the literature examined we did report this in our review. We also commented on possible causal relations between modulation of mRNA splicing and treatment efficacy, which is not always clear, because the experimental drugs may also affect other processes. We apologize if this was perhaps unnoted due to the density of our review. In many cases, we reported on mechanism and efficacy in the same short section. Examples of such sections are what we wrote on Thailanstatin A and analogues in lines 1811-1819; about SF3B1 inhibitors in lines 1849-1876; on anticancer sulfonamides in lines 1966-1979; on T-025 in lines 2060-2068; on combined E7107, EPZ015866 and MS023 in lines 2138-2152; and on palbociclib in lines 2162-2178. Indeed, in some instances we were perhaps too condense, such as e.g. for HKF in lines 1594-1598; BN82685 on lines 1645-1647; TCA on lines 1658-1661; and SRPIN340 on lines 2020-2022. Therefore, we expanded the notes on the anticancer effects of HKF (new lines 1598-1600),  BN82685 (new lines 1647-1649), TCA (new lines 1661-1664) and SRPIN340 (new lines 2022-2023) in our revised manuscript. However, we decided not to extensively elaborate on the known anticancer efficacies of these compounds in order to not make our review even longer. We think this is acceptable, because we provide the relevant references where details on the observed effects can be found. We also decided not to include known anticancer effects of compounds if they were likely not mediated via modulation of mRNA splicing. An example of this is DS-437, which has shown antitumor effects in a murine CRC model. However, this was ascribed to inhibition of FOXP3 dimethylation resulting in Treg depletion and thus activation of antitumor immunity. In such cases, we consider it not relevant to mention the anticancer efficacy of the compounds. However, some drug screens reviewed in our manuscript identified molecules that were already in use as anticancer agents before they were identified as inhibitors of mRNA splicing, such as CLK inhibitors and TOP1 inhibitors. Here, anticancer efficacy is thus evident. In those cases, we mention this and comment on if the anticancer effects are likely due to inhibition of mRNA splicing (text not changed).

We agree with the reviewer that it would be great if readers could be guided to the best current strategy. This is however in our opinion not really possible, because direct comparisons of experimental drugs targeting different components of the spliceosome in the same cancer models have not been made. To mention this, we have added the following sentences in the conclusions and future perspectives section of our revised manuscript: “Which of the many strategies to target the mRNA splicing machinery is the most promising to treat cancer is difficult to say. Side-by-side comparisons of different spliceosome-targeting drugs in preclinical models have not been made and only few experimental drugs have reached clinical investigation. However, with many drug development programs on the way, more insight into the utility of different inhibitors of the mRNA splicing machinery will be gained in the coming years.” (lines 2237-2243).

I think the title of the review could be improved to reflect the general contents of the manuscript.

Response: We agree with the reviewer. Indeed, our original title did not cover the part of our review that deals with the biology of the spliceosome. We have changed the title of our manuscript, which now reads “Biology of the mRNA splicing machinery and its dysregulation in cancer providing therapeutic opportunities.”

In the review, in different parts, it is reported that the spliceosome builds up around the intron, so giving the misconception that spliceosome is only able to define the intron. However, there are several experimental evidences that spliceosome can build up also around the exon, in particular when introns are very long. So, authors should include also this concept in the review.

Response: We agree with this remark. Indeed, we generally described the recruitment of spliceosome components to the pre-mRNA substrate with its recognition motifs for spliceosome proteins on intron-exon boundaries and within introns according to the model that is largely built upon studies done in yeast, where introns are usually short and exons long. We thank the reviewer for pointing at the concept of exon definition that was overlooked in our original manuscript. To address this, we added new text in subsection 2.1.4, where we describe the assembly of spliceosome proteins onto the pre-mRNA substrate. We introduce the subject on lines 399-407 “Notably, the genomic architecture in lower eukaryotes such as yeast is different from that in higher eukaryotes such as mammals. The former usually have relatively long exons and short introns; the latter often short exons and sometimes very long introns. Most fundamental studies into the biology of the spliceosome were done in yeast, or using recombinant transcripts with short introns. Therefore, the general description of spliceosome assembly below primarily applies to pre-mRNAs with short introns, known as the intron definition model. The steps in the process that are probably different for transcripts with long introns, according to the postulated exon definition model [91], are mentioned separately.”; we changed the text describing the recruitment of U1 and U2 snRNPs in complex E on lines 427-431, which now reads “Subsequently, on short introns U2 snRNP is recruited through interacting with U1 snRNP and SF1, replacing SF1 at the BPS. The association of U2 snRNP is further stabilized by U2AF65 [73]. On long introns, U2 snRNP is also recruited to SF1 and U2AF65 near the 3’ss and associates with U1 snRNP, but positions the U1 snRNP to the downstream 5’ss of the next intron [91].”; and added the following text to the description of complex A assembly on lines 440-454 “This is in line with the exon definition model, where the recruited U1 snRNP and U2 snRNP are to participate in splicing of different introns on either side of the exon. For splicing of transcripts with long introns, neighboring exons must be juxtaposed, existing U1 snRNP-U2 snRNP interactions across exons need to be broken; and new contacts spanning introns need to be established. This transition is still poorly understood, but the process is inhibited by hnRNPI. In the presence of hnRNPI spliceosome assembly with U1 and U2 snRNPs recruited around exons stalls in an A-like complex [99], showing that the transition occurs prior to U4/U6.U5 tri-snRNP recruitment. Recently, a model for early spliceosome assembly was proposed that unifies the intron definition and exon definition models [94]. Based on cryo-EM analysis of in vitro assembled complexes E and A it was concluded that the same structure can be formed across either an intron or an exon. Structural constraints of complexes formed across short exons make it difficult for the U4/U6.U5 tri-snRNP to subsequently join the spliceosome. This is postulated to be a main trigger for remodeling U1 snRNP-U2 snRNP interactions into an intron-spanning complex, allowing further spliceosome assembly [94].”

On page 12, line 523, the authors state that “the strength of a splice site is dependent on the gene context”. Well, while this is true when only considering the exon-intron sequence, it is well known that the “strength” of splice sites, as also reported by authors, is influenced by many factors, including polymerase speed, nucleosome position. Therefore, the authors should smooth this sentence.

Response: Indeed, as we describe in our review, splice site recognition is influenced by many factors. The cited sentence should be read in the context of its paragraph, where we added nuance to the strict definition of strong and weak splice sites  on the simple basis of their compliance to the consensus splice site sequence, introducing the observation by Wong et al showing that the same splice site sequence has a different strength when placed in a different gene. To avoid misunderstanding, we replaced the original sentence “The strength of a splice site is dependent on the gene context.” by the new sentence “However, the strength of a splice site is not only dependent on its sequence; it is also influenced by the gene context.” on lines 550-552 in our revised manuscript.

In light of point 4, the authors are forgiving to mention that also the RNA secondary structure can influence the splicing outcome, by influencing splice site recognition or binding of splicing factors.

Response: Although we briefly touched upon this aspect, but only in the context of adenosine methylation changing the mRNA secondary structure, we agree with the reviewer that this aspect was largely overlooked in our original manuscript. We have included a new section 2.2.2 in our manuscript on lines 657-687 that reads as follows “2.2.2. Effect of secondary mRNA structure

The secondary structure of the pre-mRNA substrate can also affect (alternative) splicing (reviewed in [140]). For example, stem-loop hairpin structures caused by intramolecular basepairing alter the local accessibility for trans-acting proteins. If a cis-acting enhancer or silencer motif is present in the stem of a hairpin structure, it is generally inaccessible for protein binding and thus dysfunctional for mRNA splicing regulation. Conversely, ESEs that are located immediately downstream of a hairpin structure usually exhibit strong enhancer activity. Hence, mutations in the pre-mRNA sequence that change its secondary structure by increasing or decreasing the stability of hairpins may affect splice factor binding and thus AS. In addition, mutations near splice sites may create hairpins that sequester the splice site sequence, thereby inhibiting recruitment of the spliceosome. In contrast, certain RBPs bind specifically to hairpins. For example, MBNL1 was shown to bind a hairpin sequence that contains a binding site for U2AF65 in its loop portion. MBNL1 and U2AF65 compete for binding, where MBNL1 inhibits U2AF65 binding and thus U2 snRNP recruitment when the intron adopts a hairpin structure, whereas U2AF65 binds to allow splicing when the sequence is in its single-strand fashion [141]. Another secondary RNA structure that affects splicing efficiency is the so-called G-quadruplex structure. This can be formed by a conserved sequence motif comprising at least four tracts of GG dinucleotides, folding into a helix consisting of stacked planar structures that are held together through Hoogsteen hydrogen bonding. In pull-down assays with G-quadruplex-forming RNA oligonucleotides the spliceosome proteins U2AF65, SRSF1, SRSF9, hnRNPF, hnRNPH and hnRNPU were identified [142]. The effects of G-quadruplex sequences on AS probably depends on their position within the RNA sequence and on which trans-acting protein they bind. At least for hnRNPF and hnRNPH there is suggestive evidence that their binding to G-quadruplex structures changes mRNA splicing [143,144]. Finally, RNA duplex structures formed by intramolecular interaction between sequence motifs located at sometimes very large distance in introns were found to determine AS. An example of this is the alternative exon incorporation in the FGFR2 gene causing different isoforms expressed in different cell types. The formation of the duplex structure was concluded to function solely to juxtaposition otherwise distant cis-acting elements [145].”

Authors should also include, among other splicing affecting factors, also the so call istonic code, where a different splicing outcome is observed in presence of different istonic protein modification, thus relating the splicing with chromatin structure and epigenetic features.

Response: We assume here the reviewer refers to histone modifications. This aspect was also only briefly touched upon, in the context of its effect on elongation rate (lines 714-718). We agree that we should write a little bit more about histone modification, as it also affects mRNA splicing independent of its effect on transcription rate. Therefore, we include new short section 2.2.4 in our revised manuscript on lines 736-752 that reads “”2.2.4. Effect of chromatin structure

Apart from epigenetic processes affecting mRNA splicing through the kinetic coupling of transcription and splicing discussed above, there is also evidence that chromatin organization and histone modifications have more direct effects on mRNA splicing, by contributing to exon definition and splice site choice (reviewed in [154,155]). Nucleosomes are found enriched at exon-intron junctions suggesting that they play a role in exon definition [156,157]. Moreover, in alternatively spliced genes they are more highly enriched around included exons than around excluded ones. Together, this strongly argues for a function of nucleosome positioning in regulating splicing [156]. Also certain histone modification marks are found enriched at exons, more than would be expected as a consequence of nucleosome distribution [157]. For some histone modifications a mechanism by which they modulate mRNA splicing was revealed. For example, H3K4me3 was shown to bind the U2 snRNA via the chromatin remodeling protein CHD1; and this promoted recruitment of the U2 snRNP to the pre-mRNA branch site [158]. Another example is the recruitment of hnRNPI via MRG15 binding to H3K36me3 [159]. Thus, histone modifications appear to influence AS by recruiting mRNA splicing factors to the pre-mRNA substrate via chromatin-binding adapter proteins.”

Moreover, authors should also include the importance of exonization events in presence of Ale elements, an important feature for the creation of new exons and thus the evolution of complexity.

Response: This is indeed a topic that was not discussed in our review. We thank the reviewer for pointing at this omission. In our revised manuscript, we include the following new text on lines 572-585 to discuss exonization by intronic Alu elements: “Obviously, mutations in the splice site sequences, which are associated with disease but rare in lung cancer (see section 3), affect their recognition by the mRNA splicing machinery and thus AS. In addition, mutations elsewhere on the pre-mRNA transcript may introduce cryptic splice sites that compete with the canonical sites. Here, in particular the role of Alu retrotransposons is worth mentioning. These most abundant transposable elements in the human genome are present in most primary transcripts [126]. Alu elements contain cryptic splice sites and when they are inserted in an intron in the antisense orientation, their poly(A) tract can be recognized as PPT sequence, promoting recruitment of the spliceosome to the cryptic splice site [126]. If multiple Alu elements are integrated in a long intron they can together delineate a cryptic exon. When the cryptic sites are used by the mRNA splicing machinery, this creates a new exon. This process known as exonization contributes to evolutionary complexity. In addition, a systematic analysis of Alu elements integrated in introns near exons with rather weak splice sites showed that they can alter exon incorporation efficiencies [127].” In addition, in the section on trans-acting mRNA splicing factors we inserted on lines 606-608 the new text “Interestingly, hnRNPC was also shown to protect the transcriptome against exonization by competing with U2AF65 for binding at cryptic splice sites created by integrated Alu elements [97].”

In the last section of the review, the authors report an overview of investigations to discover therapeutic targets in the spliceosome and give an overview of 18 inhibitors and modulators of the mRNA splicing process identified so far. For sake of completeness, authors should also report AON (ESSENCE, TOE) and U1snRNA based strategies for the modulation of splicing outcome, with therapeutic potential also for cancer cells.

Response: The reviewer proposes that we include a discussion on the therapeutic use of AONs and U1 snRNA based strategies; the latter presumably meaning the use of recombinant U1 snRNAs that target a specific exon or mutant 5’ss, such as those under study to target mRNA splicing defects in SMA. Correcting AS with AONs or recombinant U1 snRNAs was deliberately not discussed in our review. As stated in the introduction to section 4 on lines 1315-1319 (in our original manuscript reading: “Based on the fundamental knowledge of dysregulated mRNA splicing in cancer, there have been many efforts to correct specific AS events in cancer cells, which have been reviewed elsewhere [238]. In this review, we focus on attempts to modulate the mRNA splicing machinery.”), we decided to discuss only the efforts made to inhibit the spliceosome; not the attempts to correct aberrant AS events. One reason is because correction of AS in cancer has been reviewed recently elsewhere; another reason is that we think that targeting the spliceosome is more generally applicable and probably more effective to treat lung cancer. The latter we mention in the conclusion section (lines 2213-2222). To emphasize the focus of our review, we have slightly modified the text on lines 1315-1319, which now reads “Based on the fundamental knowledge of dysregulated mRNA splicing in cancer, there have been many efforts to correct specific AS events in cancer cells, for example using splice variant-specific siRNAs or antisense oligonucleotides. These approaches have been reviewed elsewhere [238]. In this review, we exclusively focus on attempts to modulate the mRNA splicing machinery.”

Reviewer 2 Report

The review manuscript submitted by Blijlevens et al discusses mainly two scientific concepts. One is the molecular dynamics of RNA processing (splicing) and second is its therapeutic targeting in the context of lung cancer.

This article is pretty comprehensive. Authors attempted to discuss nearly every aspect of the topic, which resulted in the 62 pages long manuscript. Many of the text in the first two sections and first five figures (focused on the molecular mechanism) is bookish and it may be way beyond the objective of this review. So, I would suggest to split this manuscript in two manuscripts for publication. Otherwise, it will be difficult to keep interest of readers till the end of review.

Author Response

This article is pretty comprehensive. Authors attempted to discuss nearly every aspect of the topic, which resulted in the 62 pages long manuscript. Many of the text in the first two sections and first five figures (focused on the molecular mechanism) is bookish and it may be way beyond the objective of this review. So, I would suggest to split this manuscript in two manuscripts for publication. Otherwise, it will be difficult to keep interest of readers till the end of review.

Response: We agree with the general observation that our manuscript is rather long. You suggested that it might be better to split our manuscript into two separate reviews. After consulting the Academic Editor, we have decided to adhere to their advice not to divide our manuscript in two separate reviews, but to add subsections to help readers navigate through the review.

Round 2

Reviewer 1 Report

Authors fulfilled all my concerns

Author Response

Authors fulfilled all my concerns.

Response: We thank the reviewer for the valuable comments that helped us improve our manuscript.

Reviewer 2 Report

Authors are suggested to significantly reduce the text in the first two sections and first five figures (focused on the molecular mechanism) if they do not want to split this manuscript.

Author Response

Authors are suggested to significantly reduce the text in the first two sections and first five figures (focused on the molecular mechanism) if they do not want to split this manuscript.

Response: After consulting the academic editor, we decided to introduce more subsections to help readers navigate through the review. We extended the mechanism section of the review to address the suggestions for improvement made by reviewer 1. In the second round revision, we decided not follow your suggestion to substantially reduce the length of the mechanism sections. In our opinion, this would lead to a less thorough discussion of the mechanism. Instead, we streamlined the mechanism sections, only deleting text that we consider less essential. This resulted in a modest reduction of the length of our manuscript (30 lines less).